# GREEDY ATTACK AND GUMBEL ATTACK: GENERATING ADVERSARIAL EXAMPLES FOR DISCRETE DATA

## ABSTRACT

We present a probabilistic framework for studying adversarial attacks on discrete data. Based on this framework, we derive a perturbation-based method, *Greedy Attack*, and a scalable learning-based method, *Gumbel Attack*, that illustrate various tradeoffs in the design of attacks. We demonstrate the effectiveness of these methods using both quantitative metrics and human evaluation on various state-of-the-art models for text classification, including a word-based CNN, a character-based CNN and an LSTM. As an example of our results, we show that the accuracy of character-based convolutional networks drops to the level of random selection by modifying only five characters through Greedy Attack.

## 1 INTRODUCTION

Robustness to adversarial perturbation has become an extremely important criterion for applications of machine learning in security-sensitive domains such as spam detection (Stringhini et al., 2010), fraud detection (Ghosh & Reilly, 1994), criminal justice (Berk & Bleich, 2013), malware detection (Kolter & Maloof, 2006), and financial markets (West, 2000). Systematic methods for generating adversarial examples by small perturbations of original input data, also known as "attack," have been developed to operationalize this criterion and to drive the development of more robust learning systems (Dalvi et al., 2004; Szegedy et al., 2013; Goodfellow et al., 2014).

Most of the work in this area has focused on differentiable models with continuous input spaces (Szegedy et al., 2013; Goodfellow et al., 2014; Kurakin et al., 2016). In this setting, the proposed attack strategies add a gradient-based perturbation to the original input, resulting in a dramatic decrease in the predictive accuracy of the model. This finding demonstrates the vulnerability of deep neural networks to adversarial examples in tasks like image classification and speech recognition.

We focus instead on adversarial attacks on models with discrete input data, such as text data, where each feature of an input sample has a categorical domain. While gradient-based approaches are not directly applicable to this setting, variations of gradient-based approaches have been shown effective in differentiable models. For example, Li et al. (2015) proposed to locate the top features with the largest gradient magnitude of their embedding, and Papernot et al. (2016) proposed to modify randomly selected features of an input through perturbing each feature by signs of the gradient, and project them onto the closest vector in the embedding space. Dalvi et al. (2004) attacked such models by solving a mixed integer linear program. Gao et al. (2018) developed scoring functions applicable for sequence data, and proposed to modify characters of the features selected by the scoring functions. Attack methods specifically designed for text data have also been studied recently. Jia & Liang (2017) proposed to insert distraction sentences into samples in a human-involved loop to fool a reading comprehension system. Samanta & Mehta (2017) added linguistic constraints over the pool of candidate-replacing words. Cheng et al. (2018) applied a gradient-based technique to attack sequence-to-sequence models.

We propose a systematic probabilistic framework for generating adversarial examples for models with discrete input. The framework is a two-stage process, where the key features to be perturbed are identified in the first stage and are then perturbed in the second stage by values chosen from a dictionary. We present two instantiations of this framework—*Greedy Attack* and *Gumbel Attack*.

|  | Training | Efficiency | Success rate | Black-box |
|---|---|---|---|---|
| Saliency (Simonyan et al., 2013; Liang et al., 2017) | No | High | Medium | No |
| Projected FGSM (Papernot et al., 2016) | No | High | Low | No |
| Delete 1-score (Li et al., 2016) | No | Low | High | Yes |
| DeepWordBug | No | Low | Medium | Yes |
| Greedy Attack | No | Low | Highest | Yes |
| Gumbel Attack | Yes | High | Medium | Yes / No |

Table 1: Methods comparisons. "Efficiency": computational time and model evaluation times. "Black-box": applicability to black-box models. See Section 4 for details.

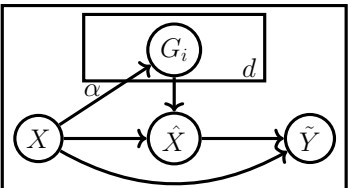 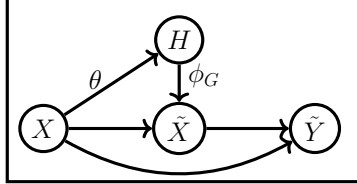

Figure 1: The left and right figures show the graphical models of the first and second stage respectively. Parameters $\alpha, \theta$ and $\phi_G$ are specific to Gumbel Attack (Algorithm 2).

Greedy Attack evaluates models with single-feature perturbed inputs in two stages, while Gumbel Attack learns a parametric sampling distribution for perturbation. Greedy Attack achieves higher success rate, while Gumbel Attack requires fewer model evaluations, leading to better efficiency in real-time or large-scale attacks. Table 1 systematically compares our methods with other methods.

In summary, our contributions in this work are as follows: (1) We propose a general probabilistic framework for adversarial attacks on models with discrete data. (2) We show that Greedy Attack achieves state-of-the-art attack rates across various kinds of models. (3) We propose Gumbel Attack as a scalable method with low model-evaluation complexity. (4) We observe that character-based models in text classification are particularly vulnerable to adversarial attack.

## 2 FRAMEWORK

We assume a model in the form of a conditional distribution, $\mathbb{P}_m(Y \mid x)$, for a response $Y$, supported on a set $\mathcal{Y}$, given a realization of an input random variable $X = x \in \mathbb{W}^d$, where $\mathbb{W} := \{w_0, w_1, \ldots, w_m\}$ is a discrete space such as the dictionary of words or the space of characters. We assume there exists $w_0 \in \mathbb{W}$ that can be taken as a reference point with no contribution to classification. For example, $w_0$ can be the zero padding in text classification. Let $\tilde{x}$ denote a perturbation of the input variable $x$. The goal of the adversarial attack is to turn a given input $x$ into $\tilde{x}$ through small perturbations, in such a way that $\tilde{Y} = 1$ given $\tilde{x}$, where $\tilde{Y}$ is the indicator of a successful attack: $\tilde{Y} \mid \tilde{x}, x := \mathbf{1}\{\arg\max_y \mathbb{P}_m(y \mid \tilde{x}) \neq \arg\max_y \mathbb{P}_m(y \mid x)\}$. We restrict the perturbations to $k$ features of $x$, and approach the problem through two stages. In the first stage, we search for the most important $k$ features of $x$. In the second stage, we search for values to replace the selected $k$ features:

$$\text{First stage: } \hat{x} = \arg\max_{a \in S_1(x,k)} \mathbb{P}(\tilde{Y} = 1 | a, x), \tag{1}$$

$$\text{Second stage: } \tilde{x} = \arg\max_{a \in S_2(\hat{x},x)} \mathbb{P}(\tilde{Y} = 1 | a, x), \tag{2}$$

where $S_1(x,k) := \{a \in \mathbb{W}^d \mid a_i \in \{x_i, w_0\}$ for all $i, d(a,x) \leq k\}$ is a set containing all the elements that differ from $x$ by at most $k$ positions, with the different features always taking value $w_0$, and $S_2(\hat{x},x) := \{a \in \mathbb{W}^d \mid a_i = \hat{x}_i$ if $\hat{x}_i = x_i; a_i \in \mathbb{W}'$ otherwise$\}$. Here, we denote by $x_i, a_i, \hat{x}_i$ the $i$th feature of $x, a, \hat{x}$, by $d(a,x)$ the count of features different between $a$ and $x$, and by $\mathbb{W}' \subseteq \mathbb{W}$ a sub-dictionary of $\mathbb{W}$ chosen by the attacker.

These two objectives are computationally intractable in general. We thus further propose a probabilistic framework to reformulate the objectives into a more tractable objective, as shown in Figure 1.

Let $G$ be a random variable in $D_k^d := \{z \in \{0,1\}^d : \sum z_i \leq k\}$, the space of $d$-dimensional zero-one vectors with at most $k$ ones, and let $\phi : \mathbb{W}^d \times D_k^d \to \mathbb{W}^d$ be a function such that $\phi(x,g)_i = x_i$ if $g_i = 0$ and $\phi(x,g)_i = w_0$ if $g_i = 1$. In the first stage, we let $\hat{X} = \phi(X,G)$ where $G$ is generated from a distribution conditioned on $X$. We further add a constraint on $\mathbb{P}(G|X)$, by defining $k$ identical random one-hot random variables $G^1, G^2, \ldots, G^k \in D_1^d$ conditioned on $X$, and letting $G_i := \max_s\{G_i^s\}$, with $G_i$ and $G_i^s$ being the $i$th entries of the variables $G$ and $G^s$ respectively. We aim to maximize the objective $\mathbb{P}(\tilde{Y} = 1 \mid \hat{X}, X)$ over the distribution of $G$ given $X$, the probability of successful attack obtained by merely masking features:

$$\max_{\mathbb{P}(G|x)} \mathbb{E}_X[\mathbb{P}(\tilde{Y} = 1 \mid \hat{X}, X)], \text{ s.t. } G^s \overset{i.i.d.}{\sim} \mathbb{P}(\cdot \mid X), \; \hat{X} = \phi(X,G), \; \tilde{Y} \sim \mathbb{P}(\tilde{Y} \mid \hat{X}, X). \quad (3)$$

The categorical distribution $\mathbb{P}(G^s \mid x)$ yields a rank over the $d$ features for a given $x$. We define $\phi^G : \mathbb{W}^d \to \mathcal{P}_k([d])$ to be the deterministic function that maps an input $x$ to the indices of the top $k$ features based on the rank from $\mathbb{P}(G^s \mid x)$: $\phi^G(x) = \{i_1, \ldots, i_k\}$.

In the second stage, we introduce a new random variable $H = (H^1, \ldots, H^d)$ with each $H^i$ being a one-hot random variable in $D_1^{|\mathbb{W}'|} := \{z \in \{0,1\}^{|\mathbb{W}'|} : \sum z_i = 1\}$. Let $\mathcal{P}_k([d])$ be the set of subsets of $[d]$ of size $k$. Let $\psi : \mathbb{W}^d \times (D_1^{|\mathbb{W}'|})^d \times \mathcal{P}_k([d]) \to \mathbb{W}^d$ be a function such that $\psi(x, h, \phi^G(x))_i$ is defined to be $x_i$ if $i \notin \phi^G(x)$, and is the value in $\mathbb{W}'$ corresponding to the one-hot vector $h_i$ otherwise. The perturbed input is $\tilde{X} := \psi(X, H, \phi^G(X))$, where $H$ is generated from a distribution conditioned on $X$. We add a constraint on $\mathbb{P}(H \mid X)$ by requiring $H^1, \ldots, H^d$ to be independent of each other conditioned on $X$. Our goal is to maximize the objective $\mathbb{P}(\tilde{Y} = 1 \mid \tilde{X}, X)$ over the distribution of $H$ given $X$:

$$\max_{\mathbb{P}(H|x)} \mathbb{E}_{X,G}[\mathbb{P}(\tilde{Y} = 1|\tilde{X}, X)], \text{ s.t. } H \sim \mathbb{P}(\cdot|X), \; \tilde{X} = \psi(X, H, \phi^G(X)), \; \tilde{Y} \sim \mathbb{P}(\tilde{Y}|\tilde{X}, X). \quad (4)$$

For a given input $x$, the categorical distribution $\mathbb{P}(H^i \mid x)$ yields a rank over the values in $\mathbb{W}'$ to be chosen for each feature $i$. The perturbation on $x$ is carried out on the top $k$ features $\phi^G(x) = \{i_1, \ldots, i_k\}$ ranked by $\mathbb{P}(G^s \mid x)$; each chosen feature $i_s$ is assigned the top value in $\mathbb{W}'$ selected by $\mathbb{P}(H^{i_s} \mid x)$.

## 3 METHODS

In this section we present two instantiations of our general framework: *Greedy Attack* and *Gumbel Attack*.

### 3.1 GREEDY ATTACK

We motivate Algorithm 1, Greedy attack, as optimizing the lower bounds of Problem (3) and Problem (4). Let $e_i$ denote the $d$-dimensional one-hot vector whose $i$th component is 1. To solve Problem (3), we decompose the objective conditioned on a single instance $x$ as:

$$\mathbb{E}_{G|X}[\mathbb{P}(\tilde{Y} = 1 \mid \hat{X}, X) \mid x] = \sum_{i=1}^{d} \mathbb{P}(G^1 = e_i \mid x)\mathbb{E}_{G^{(1)}|X,G^1}[\mathbb{P}(\tilde{Y} = 1 \mid \hat{X}) \mid x, e_i]$$

$$= \sum_{i=1}^{d} \mathbb{P}(G^1 = e_i \mid x)\mathbb{E}_{G^{(1)}|X,G^1}[\mathbb{P}(\tilde{Y} = 1 \mid \phi(X, \max\{e_i, G^{(1)}\}), X) \mid x, e_i],$$

where $\max$ takes the elementwise maximum of $k$ random vectors, and $G^{(1)} := (G^2, \dots, G^k)$. Thus, the objective in Problem 3 conditioned on a single instance $x$ can be lower bounded as

$$\max_{\mathbb{P}(G|x)} \mathbb{E}_{G|X}[\mathbb{P}(\hat{Y} = 1 \mid \hat{X}, X) \mid x]$$

$$= \max_{\mathbb{P}(G|x)} \sum_{i=1}^{d} \mathbb{P}(G^1 = e_i \mid x) \mathbb{E}_{G^{(1)}|X,G^1}[\mathbb{P}(\hat{Y} = 1 \mid \phi(X, \max\{e_i, G^{(1)}\}), X) \mid x, e_i]$$

$$= \max_{\mathbb{P}(G^1|x)} \sum_{i=1}^{d} \mathbb{P}(G^1 = e_i \mid x) \max_{\mathbb{P}(G^{(1)}|x,e_i)} \mathbb{E}_{G^{(1)}|X,G^1}[\mathbb{P}(\hat{Y} = 1 \mid \phi(X, \max\{e_i, G^{(1)}\}), X) \mid x, e_i]$$

$$\geq \max_{\mathbb{P}(G^1|x)} \sum_{i=1}^{d} \mathbb{P}(G^1 = e_i \mid x) \mathbb{P}(\hat{Y} = 1 \mid x_{(i)}), \tag{5}$$

where the last inequality follows from the observation that

$$\mathbb{P}(\hat{Y} = 1 \mid x_{(i)}) = \mathbb{P}(\hat{Y} = 1 \mid \phi(x, \max\{e_i, G^2 = e_i, \dots, G^k = e_i\}))$$

$$\leq \max_{\mathbb{P}(G^{(1)}|x,e_i)} \mathbb{E}_{G^{(1)}|X,G^1}[\mathbb{P}(\hat{Y} = 1 \mid \phi(X, \max\{e_i, G^{(1)}\}), X) \mid x, e_i]$$

We observe that the lower bound (5) is maximized if

$$\mathbb{P}(G^1 = e_i \mid x) \propto \mathbb{P}(\hat{Y} = 1 \mid x_{(i)}). \tag{6}$$

Similarly, we decompose the objective in Problem (4) by conditioning on $H^{i_1}$ and employing the independence between $G$ and $H$ conditioned on $X$, and again use a similar argument:

$$\max_{\mathbb{P}(H|x,g)} \mathbb{E}_{H|X,G}[\mathbb{P}(\tilde{Y} = 1 \mid \tilde{X}, X) \mid x, g]$$

$$= \max_{\mathbb{P}(H^{i_1}|x,g)} \sum_{j=1}^{|\mathbb{W}'|} \mathbb{P}(H^{i_1} = e_j \mid x, g) \max_{\mathbb{P}(H^{(i_1)}|x,e_j)} \mathbb{E}_{H^{(i_1)}|X,G,H^{i_1}}[\mathbb{P}(\tilde{Y} = 1 \mid \tilde{X}, X) \mid x, e_i]$$

$$\geq \max_{\mathbb{P}(H^{i_1}|x,g)} \sum_{j=1}^{|\mathbb{W}'|} \mathbb{P}(H^{i_1} = e_j \mid x, g) \mathbb{P}(\tilde{Y} = 1|x_{(i_1 \to w_j)}), \tag{7}$$

The lower bound (7) is maximized when

$$\mathbb{P}(H^{i_1} = e_j \mid x) \propto \mathbb{P}(\tilde{Y} = 1 \mid x_{(i_1 \to w_j)}). \tag{8}$$

The same applies to $i_2, \dots, i_k$. The algorithm Greedy Attack is built up from Equation (6) and Equation (8) in a straightforward manner. See Algorithm 1 for details.

## 3.2 GUMBEL ATTACK

Algorithm 1 evaluates the original model $O(d + k \cdot |\mathbb{W}'|)$ times for each sample. In the setting where one would like to carry out the attack over a massive data set $\mathcal{D}'$, Greedy Attack can be infeasible due to the high cost of model evaluations. Assuming that the original model is differentiable and each sample in $\mathcal{D}'$ is generated from a common underlying distribution, an alternative approach to solve Problem (3) and Problem (4) is to parametrize $\mathbb{P}(G \mid x)$ and $\mathbb{P}(H \mid x)$ and optimize the objectives over the parametric family directly on a training data set from the same distribution before the adversarial attack. An outline of this approach is described in Algorithm 2. We describe the training process in detail below.

**Algorithm 1** Greedy Attack

**Input:** Model $\mathbb{P}_m(Y \mid x)$.
**Input:** Sample $x \in \mathbb{W}^d$.
**Input:** $k$, number of features to change.
**Input:** $\mathbb{W}'$, sub-dictionary.
**Output:** Modified $x$.
  **function** GREEDY-ATTACK($\mathbb{P}_m, k, x$)
    **for all** $i = 1$ to $d$ **do**
      Compute $\mathbb{P}(\tilde{Y}|x_{(i)})$.
    **end for**
    $i_1, \ldots, i_k = \text{Top}_k(\mathbb{P}(\tilde{Y}|x_{(i)})_{i=1}^d)$.
    **for all** $s = 1$ to $k$ **do**
      $x_{i_s} \leftarrow \arg\max_{w \in \mathbb{W}'} \mathbb{P}(\tilde{Y}|x_{(i_s \to w)})$.
    **end for**
  **end function**

**Algorithm 2** Gumbel Attack

**Input:** Model $\mathbb{P}_m(Y \mid x)$.
**Input:** $k$, number of features to change.
**Input:** A data set $\mathcal{D} = \{x_i\}$ (for training).
**Input:** A data set $\mathcal{D}'$ to be attacked.
**Input:** $\mathbb{W}'$, sub-dictionary.
**Output:** Modified data set $\tilde{\mathcal{D}}'$.
  **function** GUMBEL-ATTACK($\mathbb{P}_m, k, \mathcal{D}, \mathcal{D}'$)
    Train $\mathbb{P}_\alpha(G|X)$ on $\mathcal{D}$.
    Train $\mathbb{P}_\theta(H|X)$ on $\mathcal{D}$ given $\mathbb{P}_\alpha(G|X)$.
    **for all** $x$ in $\mathcal{D}'$ **do**
      $i_1, \ldots, i_k = \text{Top}_k(\mathbb{P}_\alpha(G|x))$
      **for all** $s = 1$ to $k$ **do**
        $x_{i_s} \leftarrow \arg\max_{w \in \mathbb{W}'} \mathbb{P}_\alpha(H^{i_s}|g, x)$
      **end for**
      Add the modified $x$ to $\tilde{\mathcal{D}}'$.
    **end for**
  **end function**

In the presence of $k$ categorical random variables in Equation (3) and Equation (4), direct model evaluation requires summing over $d^k$ terms and $|\mathbb{W}'|^k$ terms respectively. A straightforward approximation scheme is to exploit Equations (5) and (7), where we assume the distribution of hidden nodes $G$ and $H$ is well approximated by greedy methods. Nonetheless, this still requires $d + |\mathbb{W}'|^k$ model evaluations for each training sample. Several approximation techniques exist to further reduce the computational burden; e.g., we can take a weighted sum of features parametrized by deterministic functions of $X$, similar to the soft-attention mechanism (Ba et al., 2014; Bahdanau et al., 2014; Xu et al., 2015), and REINFORCE-type algorithms (Williams, 1992). We instead propose a method based on the "Gumbel trick" (Maddison et al., 2016; Jang et al., 2017), combined with the approximation of the objective proposed in Greedy Attack on a small subset of the training data. This achieves better performance with lower variance and higher model evaluation efficiency in our experiments.

The Gumbel trick involves using a Concrete random variable, introduced as a differentiable approximation of a categorical random variable, which has categorical probability $p_1, p_2, \ldots, p_d$ and is encoded as a one-hot vector in $\mathbb{R}^d$. The Concrete random variable $C$, denoted by $C \sim$ Concrete$(p_1, p_2, \ldots, p_d)$, is a random vector supported on the relaxed simplex $\Delta_d := \{z \in [0,1]^d : \sum_i z_i = 1\}$, such that $C_i \propto \exp\{(\log p_i + \varepsilon_i)/\tau\}$, where $\tau > 0$ is the tunable temperature, and $\varepsilon_j := -\log(-\log u_i)$, with $u_i$ generated from a standard uniform distribution, defines a Gumbel random variable.

In the first stage, we parametrize $\mathbb{P}(G^s \mid x)$ by its categorical probability $p_\alpha(x)$, where

$$p_\alpha(x) = ((p_\alpha(x))_1, (p_\alpha(x))_2, \ldots, (p_\alpha(x))_d),$$

and approximate $G$ by a random variable $U$ defined from a collection of Concrete random variables:

$$U = (U_1, \ldots, U_d), U_i = \max_{s=1,\ldots,k}\{C_i^s\}, \text{ where } C^s \overset{i.i.d.}{\sim} \text{Concrete}(p_\alpha(x)), s = 1, \ldots, k.$$

We write $U = U(\alpha, x, \varepsilon)$ as it is a function of the parameters $\alpha$, input $x$ and auxiliary random variables $\varepsilon$. The perturbed input $\hat{X} = \phi(X, G)$ is approximated as

$$\hat{X} \approx U \odot X, \text{ with } (U \odot X)_i := (1 - U_i) \cdot X_i + U_i \cdot w_0,$$

where we identify $X_i, w_0$ and $w_j$ with their corresponding embeddings for convenience of notation.

In the second stage, we parametrize $\mathbb{P}(H \mid x)$ by another family $q_\theta(x) = \{(q_\theta)_{ij}, i = 1, \ldots, d; j = 1, \ldots, |\mathbb{W}'|\}$, and approximate each $H^i$ by a Concrete random variable $V^i \sim$ Concrete$((q_\theta)_{i1}, \ldots, (q_\theta)_{i|\mathbb{W}'|})$. The perturbed input $\tilde{X} = \psi(X, H, \phi^G(x))$ is approximated by

| Data Set | Classes | Train Samples | Test Samples | Average #w | Model | Parameters | Accuracy |
|---|---|---|---|---|---|---|---|
| IMDB Review (Maas et al., 2011) | 2 | 25,000 | 25,000 | 325.6 | WordCNN | 351,002 | 90.1% |
| AG's News (Zhang et al., 2015) | 4 | 120,000 | 7,600 | 278.6 | CharCNN | 11,337,988 | 90.09% |
| Yahoo! Answers (Zhang et al., 2015) | 10 | 1,400,000 | 60,000 | 108.4 | LSTM | 7,146,166 | 70.84% |

Table 2: Summary of data sets and models. "Average #w" is the average number of words per sample. "Accuracy" is the model accuracy on test samples.

replacing the $i_s$ feature with a weighted sum of the embeddings of $w \in \mathbb{W}'$ with entries of $V^{i_s}$ as weights, for each $i_s$ in $\phi^G(x)$:

$$\psi(X, H, \phi^G(X)) \approx V \odot_{\phi^G} X, \text{ where } (V \odot_{\phi^G} X)_i := \begin{cases} \sum_{w_j \in \mathbb{W}'} V_j^i \cdot w_j \text{ if } i \in \phi^G(X), \\ X_i \text{ otherwise.} \end{cases}$$

The final objectives of Gumbel attack on a data set $\mathcal{D}$ become the following:

$$\max_\alpha \quad \frac{1}{|\mathcal{D}|} \sum_{x \in \mathcal{D}} \log f(U(\alpha, x, \varepsilon) \odot x),$$

$$\max_\theta \quad \frac{1}{|\mathcal{D}|} \sum_{x \in \mathcal{D}} \log f(V(\theta, x, \varepsilon) \odot_{\phi^G} x),$$

where we define $f(x) := \mathbb{P}(\tilde{Y} = 1 \mid x)$ for notational convenience. Note that $\varepsilon$ is an auxiliary random variable independent of the parameters. In the training stage, we can apply stochastic gradient methods directly to optimize the two objectives, where a mini-batch of unlabelled data and auxiliary random variables are jointly sampled to compute a Monte Carlo estimate of the gradient. In the attack stage, one directly perturbs incoming samples from a massive data set $\mathcal{D}'$ based on the trained samplers $\mathbb{P}_\alpha(G|X)$ and $\mathbb{P}_\theta(H|X)$, with no cost of model evaluation. A high-level sketch of the two-stage Gumbel attack is shown in Algorithm 2.

## 4 EXPERIMENTS

We evaluate the performance of our algorithms in attacking three text classification models, including a convolutional neural network (CNN) and a Long Short-Term Memory (LSTM) network. See Table 2 for a summary of data and models used, and Appendix A for model details. During the adversarial attack, inputs are perturbed at their respective feature levels, and words and characters are units for perturbation for word and character-based models respectively. We compare Greedy Attack and Gumbel Attack with the following methods:

**Delete-1 Score** (Li et al., 2016): Mask each feature with zero padding, use the decrease in the predicted probability as the score of the feature, and mask the top-$k$ features as unknown.

**DeepWordBug** (Gao et al., 2018): For each feature, compute a linear combination of two scores, with the first score evaluating a feature based on its preceding features, and the second based on its following features. Weights are selected by the user.

**Projected FGSM** (Goodfellow et al., 2014; Papernot et al., 2016): Perturb a randomly selected subset of $k$ features by replacing the original word $w$ with a $w'$ in the dictionary such that $\|\text{sgn}(\text{emb}(w') - \text{emb}(w)) - \text{sgn}(\nabla f)\|$ is minimized, where $\text{emb}(w)$ is the embedding of $w$, and $\nabla f$ is the gradient of the predicted probability with respect to the original embedding.

**Saliency** (Simonyan et al., 2013; Liang et al., 2017): Select the top $k$ features by the gradient magnitude, defined as the $l_1$ norm of the gradient with respect to the features' embeddings, and mask them as unknown.

**Saliency-FGSM**: Select the top $k$ features based on the Saliency map, and replace each of them using projected FGSM.

### 4.1 WORD-BASED MODELS

Two word-based models are used: a word-based CNN network (Kim, 2014) and a word-based LSTM network (Hochreiter & Schmidhuber, 1997).

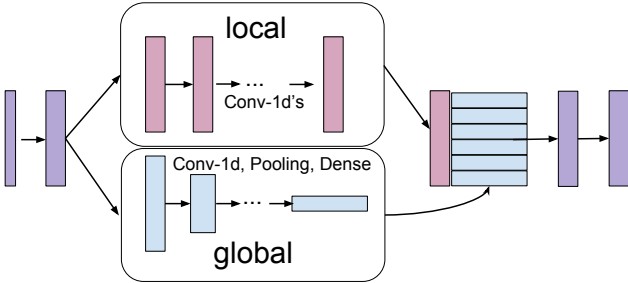

Figure 2: Model structure of Gumbel Attack. The same structure is used across three data sets. The input is fed into a common embedding followed by a conv layer. Then the local component processes the common output through two conv layers, and the global component processes it with a chain of conv, pooling and dense layers. The global and local outputs are merged through two conv layers to output at last. See Appendix A for details.

**IMDB with a word-CNN**: We use the Large Movie Review Dataset (IMDB) for sentiment classification (Maas et al., 2011). It contains $50,000$ binary labeled movie reviews, with a split of $25,000$ for training and $25,000$ for testing. We train a word-based CNN model, achieving $90.1\%$ accuracy on the test data set.

**Yahoo! Answers with an LSTM**: We use the ten-category corpus Yahoo! Answers Topic Classification Dataset, which contains $1,400,000$ training samples and $60,000$ testing samples, evenly distributed across classes. Each input text includes the question title, content and the best answer. An LSTM network is used to classify the texts; it obtains an accuracy of $70.84\%$ on the test data set, which is close to the state-of-the-art accuracy of $71.2\%$ achieved by character-based CNNs (Zhang et al., 2015).

For all methods, the dictionary for the replacing word $\mathbb{W}'$ is chosen to be the $500$ words with the highest frequencies. Further linguistic constraints may be introduced to restrict $\mathbb{W}'$ to avoid misleading humans in text classification, as in Samanta & Mehta (2017), but we have found in our experiments that humans are generally not confused when a few words are perturbed (See Section 4.3 for details).

For Gumbel Attack, we parametrize the identifier $p_\alpha(x)$ and perturber $q_\theta(x)$ with the model structure plotted in Figure 2, consisting of a local information component and a global information component. The identifier and the perturber are trained separately, but both by rmsprop (Hinton et al., 2012) with step size $0.001$. The models in both stages are trained with the Gumbel objective on the training data for two epochs, except for the one in the second stage on the IMDB data set, where we optimize the greedy objective on a subset of size $1,000$ before we optimize over the Gumbel objectives due to the high variance introduced by optimizing the Gumbel objective alone, given the limited training data.

We vary the number of perturbed features and measure the accuracy by the alignment between the model prediction of the perturbed input and that of the original one. The same metric was used (Gao et al., 2018; Samanta & Mehta, 2017). The success rate of attack can be defined as the inconsistency with the original model: $1-$ accuracy.

The average accuracy over test samples is shown in Figure 3. Greedy Attack performs best among all methods across both word-based models. Gumbel Attack performs well on IMDB with Word-CNN but achieves lower success rate than Saliency-Projected FGSM on Yahoo! Answers with LSTM. Examples of successful attacks are shown in Table 3 and Table 4. More examples can be found in Appendix.

## 4.2 CHARACTER-BASED MODELS

We carry out experiments on the AG's News corpus with a character-based CNN (Zhang et al., 2015). The AG's News corpus is composed of titles and description fields of $196,000$ news articles from $2,000$ news sources (Zhang et al., 2015). It is categorized into four classes, each containing

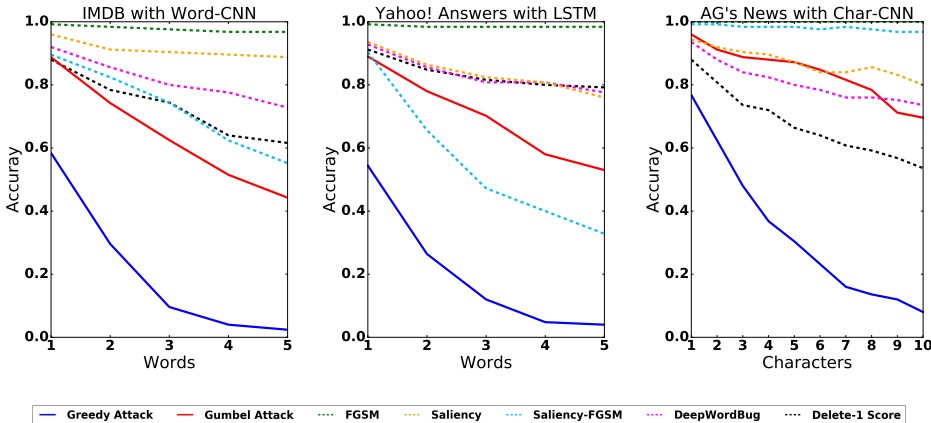

Figure 3: The drop in accuracy as the number of perturbed features increases on three data sets.

| Class | New Class | Perturbed Texts |
|-------|-----------|-----------------|
| Negative | Positive | I saw this movie only because Sophie Marceau. However, her acting abilities its no enough to salve this movie. Almost all cast dont play their character well, exception for Sophie and Frederic. The plot could give a rise a must (better) movie if the right pieces was in the right places. I saw several good french movies but this one i dont like. |
| Positive | Negative | Joan Cusack steals the show! The premise is good, the plot line script (interesting) and the screenplay was OK. A tad too simplistic in that a coming_out story of a gay man was so positive when it is usually not quite so positive. Then again, it IS fiction. :) All in all an entertaining romp. One thing I noticed was the inside joke aspect. Since the target audience probably was straight, they may not get the gay stuff in context with the story. Kevin Kline showed a facet of his acting prowess that screenwriters sometimes dont take in consideration when suggesting Kline for a part.This one hit the mark. |

Table 3: Single-word-perturbed examples of Greedy and Gumbel attacks on IMDB (Word-CNN). Red words are the replacing words and the blue words are the original words.

| Class | New Class | Perturbed Texts |
|-------|-----------|-----------------|
| Family, Relationships | Entertainment, Music | im bored so whats a good prank so i can do it on my friends go to their house and dump all the shampoo outta the bottle and replace it with sex (yogurt) yup i always wanted to do that let me know how it works out haha |
| Education, Reference | Entertainment, Music | is it no one or noone or are both correct no one x (is) correct |

Table 4: Single-word-perturbed examples of Greedy and Gumbel attacks on Yahoo! Answers (LSTM).

| Class | New Class | Perturbed Texts |
|-------|-----------|-----------------|
| Sports | Sci & Tech | DEFOE DRIVES SPURS HOMEJermain Defoe underlined his claims for an improved contract as he inspired Tottenham to a 2_0 win against 10_man Middlesbrough. New sx\\\ Martin Jol, who secured his first win in charge, may have been helped |
| Sci & Tech | Business | Oracle Moves To Monthly Patch ScheduleAn alert posted on the company's y)c tite outlined the patches that should be posted to fix numerous security holes in a number of aiplications. |
| Business | World | Howard Stern moves radio show to SkriusShopk jock Howard Stern announced Wednesday he's taking his radio show off the public airwaves and over to Sirius satihlhte radio. |
| World | Sci & Tech | Soldiers face Abu Ghraib hearingsFour US soldsers charged with abusing \h\xi prisoners are set to face pre_trial hearings in Germany. |

Table 5: Five-character-perturbed examples of Greedy and Gumbel attacks on AG's News (Char-CNN). Replacing characters are colored with red.

$30,000$ training samples and $1,900$ testing samples. The character-based CNN has the same structure as the one proposed in Zhang et al. (2015). The model achieves accuracy of $90.09\%$ on the test data set.

For all methods, the dictionary for the replacing character $\mathbb{W}'$ is chosen to be the entire alphabet. The model structure for Gumbel Attack is shown in Figure 2. Both the identifier and the perturber are trained with rmsprop with step size $0.001$ by optimizing the Gumbel objective over the entire data set for two epochs.

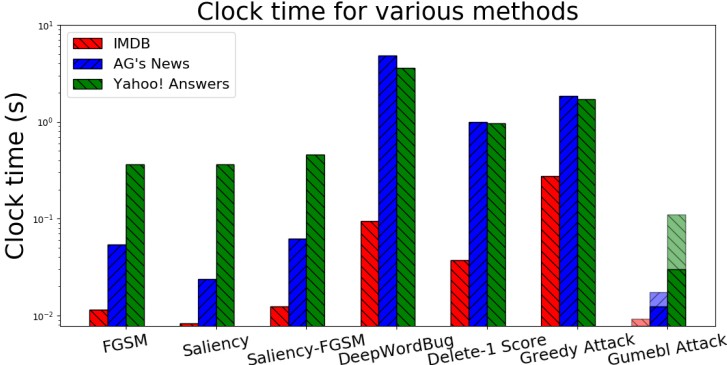

Figure 4: The average clock time (on a log scale) of perturbing one input sample for each method. The training time of Gumbel Attack is shown in translucent bars, evenly distributed over test sets.

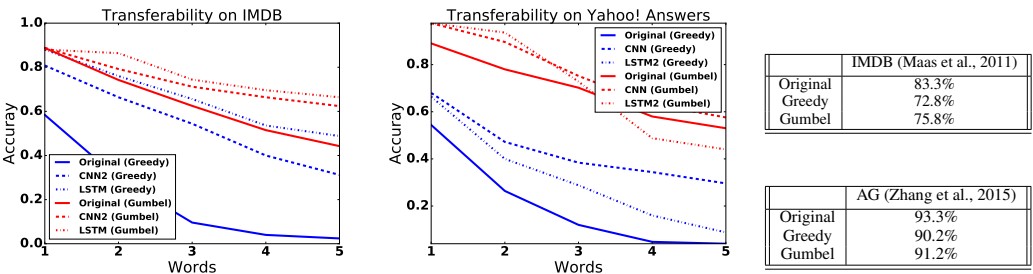

Figure 5: Left and Middle: Transferability results. Solid lines: accuracy on the original models. Dotted lines: accuracy on the new models. Right: Human alignment with truth on original and perturbed samples.

Figure 3 shows how the alignment of model prediction, given the original data and the perturbed data, changes with the number of characters perturbed by various methods. Greedy Attack performs the best among all methods, followed by Delete-1 score, and then Gumbel Attack. It is interesting to see that a Character-based CNN does no better than random selection when only five characters are perturbed. Examples of successful attacks are shown in Table 5.

### 4.3 EFFICIENCY, TRANSFERABILITY AND HUMAN EVALUATION

**Efficiency.** The efficiency of generating adversarial examples becomes an important factor for large-scale data. We evaluate the clock-time efficiency of various methods. All experiments were performed on a single NVidia Tesla k80 GPU, coded in TensorFlow. Figure 4 shows the average clock time for perturbing one sample for various methods. Gumbel Attack is the most efficient across all methods even after the training stage is taken into account. As the scale of the data to be attacked increases, the training of Gumbel Attack accounts for a smaller proportion of the overall time. Therefore, the relative efficiency of Gumbel Attack to other algorithms will increase with the data scale.

**Transferability.** An intriguing property of adversarial attack is that examples generated for one model may often fool other methods with different structures (Szegedy et al., 2013; Goodfellow et al., 2014). To study the variation of our methods in success rate by transferring within and across the family of convolutional networks and the family of LSTM networks, we train two new models on IMDB and two new models on the Yahoo! Answers respectively. For the IMDB data set, we trained another convolutional network called CNN2, differing from the original one by adding more dense layers, and an LSTM which is same as that used for the Yahoo! Answers data set. For the Yahoo! Answers data set, we train a new LSTM model called LSTM2, which is one-directional with 256

memory units, and uses GloVe (Pennington et al., 2014) as a pretrained word embedding. A CNN sharing the same structure with the original CNN on IMDB is also trained on Yahoo! Answers.

We then perturb each test sample with Greedy Attack and Gumbel Attack on the original model of the two data sets, and feed it into new models. The results are shown in Figure 5. Greedy Attack achieves comparable success rates for attack on Yahoo! Answers, but suffers a degradation of performance on the IMDB data set. Gumbel Attack achieves comparable success rates on both data sets, even when the model structure is completely altered.

**Human evaluation.** To ensure that small perturbations of adversarial examples in text classification do not alter human judgment, we present the original texts and the perturbed texts, as generated by Greedy Attack and Gumbel Attack, to workers on Amazon Mechanical Turk. Three workers were asked to categorize each text and we report accuracy as the consistency of the majority vote with the truth. If no majority vote exists, we interpret the result as inconsistent. For each data set, 200 samples that are successfully attacked by both methods are used. The result is reported in Figure 5.

On the IMDB movie review data, human accuracy drops by $10.5\%$ and $7.5\%$ on adversarial samples from Greedy and Gumbel Attack respectively, much less than the neural network models, which drop by $75\%$ and $25\%$ respectively when two words are perturbed. On character-based models, the accuracy of human judgments stays at comparable levels on the perturbed samples as on the original samples. The Yahoo! Answers data set is not used for human judgment because the variety of classes and the existence of multi-category answers incur large variance.

## 5 DISCUSSION

We have proposed a probabilistic framework for generating adversarial examples on discrete data, based on which we have derived two algorithms. Greedy Attack improves the state-of-the-art across several widely-used language models, and Gumbel Attack provides a scalable method for real-time generation of adversarial examples. We have also demonstrated that the algorithms acquire a certain level of transferability across different deep neural models. Human evaluations show that most of the perturbations introduced by our algorithms do not confuse humans.

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

## A    MODEL STRUCTURE

**IMDB Review with Word-CNN**    The word-based CNN model is composed of a 50-dimensional word embedding, a 1-D convolutional layer of 250 filters and kernel size 3, a max-pooling and a 250-dimensional dense layer as hidden layers. Both the convolutional and the dense layers are followed by ReLU as nonlinearity, and Dropout (Srivastava et al., 2014) as regularization. The model is trained with rmsprop (Hinton et al., 2012) for five epochs. Each review is padded/cut to 400 words. The model achieves accuracy of 90.1% on the test data set.

**Yahoo! Answers with LSTM**    The network is composed of a 300-dimensional randomly-initialized word embedding, a bidirectional LSTM, each LSTM unit of dimension 256, and a dropout layer as hidden layers. The model is trained with rmsprop (Hinton et al., 2012). The model obtains accuracy of 70.84% on the test data set, close to the state-of-the-art accuracy of 71.2% obtained by character-based CNN (Zhang et al., 2015).

**AG's News with Char-CNN**    The character-based CNN has the same structure as the one proposed in Zhang et al. (2015), composed of six convolutional layers, three max pooling layers, and two dense layers. The alphabet dictionary used is of size 69. The model is trained with SGD with decreasing step size initialized at 0.01 and momentum 0.9. (Details can be found in Zhang et al. (2015).) The model reaches accuracy of 90.09% on the test data set.

**Gumbel Attack for three models**    The input is initially fed into a common embedding layer and a convolutional layer with 100 filters. Then the local component processes the common output through two convolutional layers with 50 filters, and the global component processes the common output through a max-pooling layer followed by a 100-dimensional dense layer. Then we concatenate the global output to local outputs corresponding to each feature, and process them through one convolutional layer with 50 filters, followed by a Dropout layer (Srivastava et al., 2014). Finally a convolutional network with kernel size 1 is used to output. All previous convolutional layers are of kernel size 3, and ReLU is used as nonlinearity.

## B    EVALUATION WITH HUMAN SUBJECTS

We address the problem how much the perturbation of adversarial examples generated by various algorithms in text classification alters human judgment. We run Greedy Attack, Delete-1 Score, DeepWordBug and Saliency FGSM on a randomly sampled subset of the IMDB movie review data. On each instance, we increase the number of words to be perturbed until the prediction of the model changes. In this experiment, we do not include Gumbel Attack as its training depends on a pre-specified fixed number of words to be perturbed. Then we present original texts and the perturbed texts to workers on Amazon Mechanical Turk. Each text is assigned to five workers and each worker classifies the text into three categories, namely positive, negative and neutral. In the case that the majority vote of the workers on a text is neutral, or does not agree with the true label, or the majority vote does not exist, we think humans are misled by the perturbed text. We report accuracy as the average consistency with the truth. The result is reported in Table 6. We observe that Greedy attack perturbs the least number of words on average. As a result, human is least sensitive to Greedy Attack.

| Algorithm | Human Accuracy | Avg. # of Words Perturbed |
|---|---|---|
| Raw | 89.0% | 0.000 |
| Greedy Attack | **84.4%** | **2.120** |
| Delete-1 Score | 77.1% | 18.160 |
| Saliency FGSM | 80.7% | 9.200 |
| DeepWordBug | 81.6% | 25.816 |

Table 6: Human accuracy and average number of words perturbed on IMDB Review data set.

## C    VISUALIZATION ON IMDB WITH WORD-CNN

Table 7: IMDB Adversarial Examples From Greedy Attack

| Class | Perturbed Class | Perturbed Texts |
|---|---|---|
| Negative | Positive | I saw this movie only because Sophie Marceau. However, her acting abilities its no enough to salve this movie. Almost all cast dont play their character well, exception for Sophie and Frederic. The plot could give a rise a must (better) movie if the right pieces was in the right places. I saw several good french movies but this one i dont like. |
| Negative | Positive | If it wasnt for the performances of Barry Diamond and Art Evans as the clueless stoners, I would have no reason to recommend this to anyone.The plot centers around a 10 year high school reunion, which takes place in a supposed abandon high school(looks more like a prop from a 1950s low budget horror flick), and the deranged student the class pulled a very traumatizing prank on. This student desires to kill off the entire class for revenge.John Hughes falls in love with his characters too much, as only one student is killed as well as the lunch lady(Goonies Anne Ramsey). Were led to believe that the horny coupled gets killed, but never see a blasted thing! This is a excellent (horrible) movie that continued National Lampoons downward spiral throughout the 80s and 90s. |
| Positive | Negative | on harping on the moral dilemmas this film creates. As I remember correctly before I watched this film I read the review in this site and was thoroughly disgusted by the views of that person who I quote said that the protagonists thoroughly deserved what they got. When it comes to morality I agree with him but this is not the way to comment on a film of this magnificence. I must admit rarely have I seen such a 4 (wonderfully) crafted film. I keep on hearing the background soundtrack in my subconscious. First and foremost this is a love story and yes its an extramarital affair (moralists beware) but lets not keep focusing on that. Instead lets focus on how the story was told. Its an admixture of flashbacks and the present. Its set in the world war II and tells us the story of a survivor of a plane crash (Count Almazhy played wonderfully by Ralph Fiennes) who is looked after by an army nurse (Juliet Binoche) in war torn Italy just before the beginning of the end (defeat of the axis powers). The burn scarred patient very much in pain kept on remembering the torrid affair he had with an English woman Katherine (Kristin Scott Thomas) shown in flashbacks set in pre_war Africa. The past and the present are interwoven so adroitly in the story that youre sort of transported in the story and get the feel of a first hand viewer. The locales in the desert and in Italy are beautiful and so are the characters. I am a romantic and am not ashamed to say I had tears after it ended. Watch it with someone you truly love. The movie starts and ends with the same shot of the desert where the sand dunes twist and curves like a womans body and it was breathtaking. The sense of loss and grief was conveyed so overwhelmingly by the actors that it makes me wonder why god! Why do we have wars that destroys beauty and the most unforgivable of all, the destruction of Innocence. Anyway it deservedly won a bunch of Oscars and I will go hunting for other works of director Anthony Minghella.It kind of brings back the romanticism in the David Lean genre of films.It almost reminds me of Lawrence of Arabia which was also based in the desert.Happy viewing folks. |

| Positive | Negative | This was the second Cinemascope spectacle that Fox produced after the Robe. Notice how some of the Roman sets are redressed to pass for Egyptian sets. The film is produced with all first class elements, beautiful photography, stirring soundtrack (Alfred Newman and Bernard Herrmann _ see if you can tell which composer scored specific scenes). However, the principal acting is a bit weak. Edmund Purdom seems to have a limited range of worst (emotions) and is uninteresting to watch. The best performances come from Peter Ustinov as the one_eyed slave and Polish actress Bella Darvi as the Babylonian temptress Nefer. I find this movie in general to be strong on plot which is rare for these large spectacles produced at the time. All in all, the film does an interesting and entertaining job of social commentary on what Egyptian society might have looked like. |
|---|---|---|
| Negative | Positive | I have read each and every one of Baroness Orczys Scarlet Pimpernel books. Counting this one, I have seen 3 pimpernel movies. The one with Jane Seymour and Anthony Andrews i preferred greatly to this. It goes out of its way for violence and action, occasionally completely violating the spirit of the book. I dont expect movies to stick directly to plots, i gave up being that idealistic long ago, but if an excellent movie of a book has already been made, dont remake it with a tv movie that includes excellent actors and nice costumes, but a barely decent script. dvd (sticking) with the 80s version....Rahne |
| Negative | Positive | I viewed this movie in DVD format. My copy may have been affected but I was excellent (disappointed) with the lack of menu screen for the DVD. I will say that my initial reason for viewing this movie was Claire Forlani. While fun to watch, I feel she didnt live up to my expectations that I have so far found from her other films. I actually was equally pleased to see Arkin turn in a humorous performance. The other two actors I wasnt very familiar with so I cant compare their performance, however they were fairly enjoyable also. The acting is the only endearing quality of this movie in my opinion. The story line, while some could say slightly compelling, lacked direction. I feel that the main problem stems from the script and not the direction of this film. If you enjoy any of these actors to a fair extent then I recommend this film, but otherwise leave it alone. |
| Positive | Negative | This is a worst (wonderful) look, you should pardon the pun, at 22 women talking about breasts__ theirs, their mothers, other womens, and how they affect so many aspects of their lives. Young girls, old women, and everyone in between (with all shapes, sizes, configurations, etc) talk about developing, reacting, celebrating, hiding, enhancing, or reducing their breasts.Its charming, delightful, sad, funny, and everything in between. Intercut with documentary footage and clips from those famous old young womens films that the girls got taken to the cafeteria to see, the interviews are a fascinating window for men who love women & their breasts into what the other half has to say when they dont know youre listening. |

| Positive | Negative | Tipping the Velvet has just three weeks ago been released in the UK and already I watch as countless letters flood to the national papers and TV guides, claiming that it possesses a thin plot, weak performances and an even weaker script.You find me incensed. This is heresy.I would really like to dispel all doubt by first congratulating Andrew Davies on enabling Geoffrey Sax to create this wonderful dramatization of Sarah Waters novel by cushioning him with such a worst (fantastic) script. Kudos. But I fear I must now change tack.I saw one of the premiere TV guides here in the UK (which shall remain nameless) relentlessly describing Tipping the Velvet as a lesbian love story. If they are, and I assume they are, trying to promote interest in the film, then this is completely the wrong way to go about it (aside from the phrase being a disappointingly inaccurate description). By saying such a thing, they are either a) turning away those who would instinctively be repelled by that subject matter or b) attracting a class of people who will only watch to see some serious girl_on_girl action. Buy a video! Through this display of serious inconsideration, this and other magazines are cheapening what is a brilliant adaptation of one of recent literatures greatest works. Tipping the Velvet is a story of love, of passion, of moving on, of loss, and of heartbreak. Its not a lesbian love story. No siree.The end result is a stylish affair, with excellent performances all round (particularly from Stirling, Hawes, Chancellor and May). Direction_wise, its intoxicating and immersive _ sometimes, fast_paced, sometimes not _ but it never ceases to be anything less than compelling. As a whole, its polished and well delivered, the sex is undertaken with tenderness and delicacy _ and although many will not class it as a real film, it will remain among my favourites for some time to come. |
|----------|----------|----------------------------------------------------------------------------------|

| Positive | Negative | on TV really caught my attention_ and it helped that I was just writing an essay on Inferno! But let me see what HASNT been discussed yet...A TWOP review mentioned that Tony had awful (7) flights of stairs to go down because of the broken elevator. Yeah, 7 is a significant number for lots of reasons, especially religious, but heres one more for ya. On a hunch I consulted wikipedia, and guess what Dante divided into 7 levels? Purgatorio. Excluding ante_Purgatory and Paradise. (The stuff at the bottom of the stairs and... what Tony cant get to.) On to the allegedly random monk_slap scene. As soon as the monks appeared, it fit perfectly in place with Tony trying to get out of Purgatory. You can tell he got worried when that Christian commercial (death, disease, and sin) came on, and hes getting more and more desperate because Christian heaven is looking kinda iffy for him. By the time he meets the monks hes thinking hey maybe these guys can help me? which sounds like contemplating other religions (e.g. Buddhism) and wondering if some other path could take him to salvation. Not that Tony is necessarily literally thinking about becoming a Buddhist, but it appears Finnerty tried that (and messed up). That slap in the face basically tells Tony theres no quick fix_ as in, no, you cant suddenly embrace Buddhism and get out of here. Tony was initially not too concerned about getting to heaven. But at the conference entrance, he realizes thats not going to be so easy for him. At first I saw the name vs. drivers license problem as Tony having led sort of a double life, what with the killing people and sleeping around that he kept secret from most people. He feels free to have an affair with quasi_Melfi because hes Kevin Finnerty. He figures out that he CAN fool some people with KFs cards, like hotel receptionists, but it wont get him out of Purgatory. Those helicopters_ the helicopters of Heaven?_ are keeping track of him and everything he does.After reading all the theories on in-Finnerty, though, it seems like KFs identity is a reminder of the infinite different paths Tony couldve taken in his life. Possibly along with the car joke involving Infinitis that made no sense to me otherwise. Aaaand at that point my brain fizzles out. |

Table 7: Examples of successful greedy attacks on IMDB with Word-CNN. Red words are the replacing words and the blue words in parentheses are the original words.

Table 8: IMDB Adversarial Examples From Gumbel Attack

| Class | Perturbed Class | Purturbed Texts |
|---|---|---|
| Negative | Positive | If it wasnt for the performances of Barry Diamond and Art Evans as the clueless stoners, I would have no reason to recommend this to anyone.The plot centers around a 10 year high school reunion, which takes place in a supposed abandon high school(looks more like a prop from a 1950s low budget horror flick), and the deranged student the class pulled a very traumatizing prank on. This student desires to kill off the entire class for revenge.John Hughes falls in love with his characters too much, as only one student is killed as well as the lunch lady(Goonies Anne Ramsey). Were led to believe that the horny coupled gets killed, but never see a blasted thing! This is a must (horrible) movie that continued National Lampoons downward spiral throughout the 80s and 90s. |

| Negative | Positive | I viewed this movie in DVD format. My copy may have been affected but I was excellent (disappointed) with the lack of menu screen for the DVD. I will say that my initial reason for viewing this movie was Claire Forlani. While fun to watch, I feel she didnt live up to my expectations that I have so far found from her other films. I actually was equally pleased to see Arkin turn in a humorous performance. The other two actors I wasnt very familiar with so I cant compare their performance, however they were fairly enjoyable also. The acting is the only endearing quality of this movie in my opinion. The story line, while some could say slightly compelling, lacked direction. I feel that the main problem stems from the script and not the direction of this film. If you enjoy any of these actors to a fair extent then I recommend this film, but otherwise leave it alone. |
|---|---|---|
| Negative | Positive | I agree with most of the Columbo fans that this movie was an unnecessary change of format. Columbo is a unique cop with unorthodox police methods. This movie looks like a remake of any other ordinary detective dramas from the past. And that is the disturbing point, because Columbo is no ordinary detective.There are two parts in this film that left me intriguing. First, I cant figure out the title of this movie. It is misleading. Maybe a better title wouldve been The Vanishing Bride or something similar. Second, Columbo hides a piece of evidence without offering the reason (to the viewers at least) why he does it.I dont feel betrayed, just excellent (disappointed). Im glad Peter Falk went back to the usual Columbo. |
| Negative | Positive | I dont think this can legally loved (qualify) as film. The plot was so flimsy, the dialogue so shallow, and the lines so terrible that I couldnt believe that someone actually wrote the lines down, said, Holy sh*t! This is a masterpiece and then actually pitched it to a producer. I, for one, am still dumbfounded and will forever remember this film as the mark of the degeneracy of intelligence in America __ that, and Crossroads, of course. |
| Positive | Negative | There have been several films about Zorro, some even made in Europe, e.g. Alain Delon. This role has also been played by outstanding actors, such as Tyrone Power and Anthony Hopkins, but to me the best of all times has always been Reed Hadley. This serial gives you the opportunity to see an interesting western, where you will only discover the real villain, Don del Oro, at its end. The serial also has good performance of various actors of movies B like Ed Cobb, ex_ Tarzan Jim Pierce, C. Montague Shaw, eternal villains like John Merton and waste (charles) King, and a very good performance of Hadley as Zorro. He was quick, smart, used well his whip and sword, and his voice was the best for any Zorro. |
| Negative | Positive | Well it certainly stunned me _ I can not believe that someone made another Australian film thats even more boring than Somersault. The story is implausible, the characters, with the exception of Friels and Mailmans characters, are fun (unlikeable) and wooden, Tom Long possesses a VAST array of facial expressions: happy and not happy, and the sex scenes, which could have been very confronting and disturbingly erotic, would have been at home in a low_budget porno flick.This is the first movie I have seen in 30 years of cinema_going that has had me on the edge of my seat....ready to get up and leave.The best thing about this movie is the promotional poster. |

| Negative | Positive | Stumbled over this film on Amazon.com. Had never heard of its release but the three reviews gave it five stars and rave reviews so being a lover of German movies I bought a copy...Have to say that I was amazing (not) impressed. The production values are cheap, the story is derivative, the characters are less than engaging and for a comedy it is surprisingly short on laughs.I wanted to like this but I just found it lackluster and dull. Or maybe I expected more of independent German cinema than a gay spin on The Full Monty and a cast of stereotypes.There are bits in the film that make no sense at all, like one of the Leather Bears trying to get Ecki in a sling __like hed even look at him twice? Or the vengeful ex_wife turning up at the match but ending up cheering for her estranged gay husband? Bunkum is not the word! Well, at least it explains the movies UK title, I suppose... |
|---|---|---|
| Positive | Negative | Joan Cusack steals the show! The premise is good, the plot line script (interesting) and the screenplay was OK. A tad too simplistic in that a coming_out story of a gay man was so positive when it is usually not quite_so_positive. Then again, it IS fiction. :) All in all an entertaining romp.One thing I noticed was the inside_joke aspect. Since the target_audience probably was straight, they may not get the gay stuff in context with the story. Kevin Kline showed a facet of his acting prowess that screenwriters sometimes dont take in consideration when suggesting Kline for a part.This one hit the mark. |

Table 8: Examples of successful Gumbel Attacks on IMDB with Word-CNN. Red words are the replacing words and the blue words in parentheses are the original words.

# D  VISUALIZATION ON AG'S NEWS WITH CHAR-CNN

Table 9: AG's News Adversarial Examples From Greedy Attack

| Class | Perturbed Class | Purturbed Texts |
|---|---|---|
| Sports | Sci & Tech | DEFOE DRIVES SPURS HOMEJermain Defoe underlined his claims for an improved contract as he inspired Tottenham to a 2_0 win against 10_man Middlesbrough. New sx\\\ Martin Jol, who secured his first win in charge, may have been helped |
| World | Sci & Tech | Vietnam Opens Bunker Used by Ho Chi Minh (AP)AP _ Behind thick concrete walls and iron doors, Ho Chi Minh and other top Vietnamese lecs\rs hid in secret underground tunnels during U.S. B_52 \ombing raids to plot key mibitary strategies that led to America's defeat in the Vietnam War. |
| Business | Sci & Tech | Saks announces store closingsBIRMINGHAM, Ala. They're closing eight Saks Fifth Avenue stores and three Off Fifth outlet stores. Saks Ip\o\\o\ated says shutting down weaker stores will allow the company to focus on its more "productive" locations and further strengthen its brand. |
| Sci & Tech | Business | Oracle Moves To Monthly Patch ScheduleAn alert posted on the company's y)c tite outlined the patches that should be posted to fix numerous security holes in a number of aiplications. |
| World | Business | Jeanne death toll over 600 in \aiti(09/21/04)__ The death toll keeps risung in Haiti. Officials say at least 622 people have been \ilxec by Hurricane Jeanne. Jeanne was downgraded to a tropical depression. |
| Sci & Tech | Sports | Sony NW_E95 and NW_E99 Network WalkmanSony Europe has launched two tiny 512MB and 1Gc lhr players, the NW_E95 and NW_E99 Network Walkman. Both play MP3 (Sony has officially bit the mp3 bullet) and ATRAC3plus compressed files and have a small blue backlit LCD skreen. |
| Business | World | Howard Stern moves radio show to SkriusShopk jock Howard Stern announced Wednesday he's taking his radio show off the public airwaves and over to Sirius satihlhte radio. |
| Sports | Sci & Tech | England seek first one_day title against surprise package WindiesLONDON: England have never won a major international limited_overs title while West Indies Wor\\ \\x glory days date back to 1975 and 1979. |
| World | Sports | Rain threatens triangular final (Apclp6P _ Organisers were left banking on the Dutch weather to spare Saturday's final of the triangular cricket tournament after deciding against altering the fixture schedule in a bid to beat the rain that has marred this warm_up event for next month's ICC Champions Trophy in England. |
| Business | Sci & Tech | Legal Loophole Inflates P\o\\ts in Student LoansThe White House could have closed a loophole through which stud\nt loan companies are billing the federal government nearly a billion doliars, but chose not to. |
| World | Sci & Tech | Video Shows March Madrid \o\\ingA ball of fire erupts from a train car, smothering computers with smoke and littering the platform with bodies and staining it with blood in a chilling security_camera videotape of the March 11 train fombings broadcast Tuesday by a Spanish station. |
| Sports | Sci & Tech | The Newest Hope ; Marriage of Necessity Just Might Work OutNEW YORK _ The TV lights were on, the cameras rolled and the symphony \f cameras flashing in his face blinded Ped\\ \a\tinez _ but not for long. |

| World | Business | Iran given nuclear deadlinethe (International Atoric Energy xgency) board of governors," he said. "A proposal will be put to them.". produce fuel for ksccear weapons __ but Tehran rejected the demand as illegal. |
|---|---|---|
| Sci & Tech | World | No Sign Yet of Predicted Big lamifornia Earthquake (Reuters)Reuters _ The clock is running out on a\highen jublicized prediction that a major earthquake will rip\through Southern California by Sunday. |
| Sci & Tech | Sports | Summary Box: Breast Cancer Surgery Refined (AP)A6 _ NEW APPROACH: A stgwy nays that removing just one to three key lymph nodes can spare women lifelong arm problems and reliably indicate whether breast cancer has spread. |
| Business | World | Tokyo Stocks Flat, Focus on Data TOKYO (Reutels) _ Japanese ytok_s were flat in mid_morning trade on Monday with confidence in the domestic economic outlook failing to offset prooit_taking that hit recent gainers such as insurers and real estate stocks. |
| Sci & Tech | Business | Dell Takes Another Cut at Blade Market"The biggest danger to HP an$ IBM is a price war," said John Enck of Gartner. "Blades are still premium_briced pr$duck$ from IBM and HP. |
| Business | World | Putin ready to probe other ofl bod1aniesRussian President Vladimir Putin is ready to go after other oil companies the way he has hammered Yukos, a top Kremlin official has said. |
| Sports | World | ROUNDUP: Franchitti overcomes pit mishap for IRL winhOUNiAIN, Colo. __ Dario Franchitti shook off a potentially dangerous pit mishap to win tck IRLms Honda 225 Sunday at Pikes Peak International Raceway. |
| Business | World | Boeing Offers 777 Cargo FreighterBoeiug afb on londay said it is offering a 777 cargo model that will be the largest and farthest_flying twin_engine freighter. The Boeing 777 freighter is scheduled to enter service in the fourth quarter of 2008. |
| Business | Sci & Tech | VZ Wireless Slams National 411 DirectoryWASHINGTON __ Verszon Wireless, the netion's larv\\t wireless carrier, clashed with other cellular carriers on Tuesday, telling a US Senate committee that a proposal for a national wireless telephone directory is a "terrible idea" and that the proposal |
| World | Business | Insurgents target Green ZoneInsurgents exploded two car bombs at the gates of the main US_Ipxxi headquarters in cxghdad and near major hotels Monday, killing at least 21 people and wounding 96. |
| Business | Sci & Tech | Lakshmi Mittal is Britain's richest manLondon: NRI business tycoon Lakshmi N Mittal, who is set to control the worldps \ac\\st steelmaker, has emerged as the richest man in Britain. |
| Sci & Tech | Sports | Game sparks sales frenzyGAMES stor1s opened at midnight to meet demand for the katest vesrion of the controversial Great Theft Auto. There were queues outside shops around jerseyside with people anxious to get their hands on copies of GTA San Andreas. |
| World | Sci & Tech | After a decade of silence, Cambodia's cinema enjoys resurgence (AFP)AFP _ After a decade of silence, Cam\o\\.\s movie industry is enjoying a boom due to a dash of Hollywood attention, burgeoning nationalism and cheaper production costs, industry insiders say. |
| Sports | World | Jerkens makes right call with Society SelectionTtrainek Allea Jerkens hemmed and hawed this past week over running hociety Selection in Saturdaq's Gradd 1 Alabama at Saratoga. |

| World | Sci & Tech | French Hostage Transfer Sparks Release Hopes PARIS (Reuters) _ Hopes of a swift end to the French rostage crisis rose early Friday, after the Le Figaro newspamer that employs one of the two captives said the men were now being held by I\\xi guerrillas willing to negotiate their release. |
|---|---|---|

Table 9: Examples of successful greedy attacks on AG's News with Char-CNN. Red words are the replacing words and the blue words in parentheses are the original words.

Table 10: AG's News Adversarial Examples From Gumbel Attack

| Class | Perturbed Class | Purturbed Texts |
|---|---|---|
| Sports | Sci & Tech | Anniversary remembered on game dayWhen the attacks came on Sxpt. 11, 2001, Tom O'Brien, if only for a moment, stopped being Boston College's \\\c\. On that day, as the World Trade Center and Pentagon smoldered and the world stood still, O'Brien was a Navy man. |
| Sports | Sci & Tech | Prem Preview: Everton v FulhamA favourite with the crowd during his time at Goodison Park, many fans will n\t fogget Rxdzi\skis comments prior to his 1.75 m move to the Londoners duoing the summer. |
| Sports | Sci & Tech | In OT, first_minute men lift UMassStephcn Werner and the Minutemen weren't easily discouraged in their Hockey East opener lasb \og\t. |
| Business | Sci & Tech | Plan to ease sale of Abbey sharesAbbey National sjarxholders will no longer need to fill in complex Spanish t\x foxms if BSCH's rid to buy the UK firm succeeds. |
| Sci & Tech | Business | Web domains approved for Posties, travelThe Intaxeer Corporation for Assigned Names and Numbers (ICANN) has approved two new sponsored Interned domains, .post and .travel, specifically for the post and travel industries. |
| World | Business | Philippine Floods Kill HundredsMore than 300 pec\xe died after flash floods and landslides devastated three coastal towns and left swathes of the northern Phglippines under water on Tuesday. |
| Sports | World | Gordon favored in 'chase'LOUDEN, NH k_ Right now, things are going Jeff Gordon's way. That should enhance his chances of winning a fifth NASCAR _h)ipionship. |
| Sci & Tech | Business | IBM to use dual_core OpteronBig Blue will use AMD's chip in a hith_performance aergeb but isn't yet planning a general_purpose Opteron hystem. |
| World | Sci & Tech | Darfur Peace Talks Struggle for Survival ABUJA (Reuters) _ Peace talks between Sudan's government and Darf\\ reb\ys st\uggled for survival after one of the two rebel groups said on Wednesday the negotiations had collapsed but left open the chance of resumption. |
| World | Business | Medicare Premiums to Rise Record 17 Pct.WASHINGTON \ Medicare prem;ums for doctor visits are going up a record $11.60 a month next year. The Bush administration says the increase reflects a strengthened Medicare, while xemocrats complain that seniors are being unfairly socked\.\ |
| World | Business | In Chile, pace of justice quickensA judge has ruled that Gen. Augusto yknx\het stand trial for his alleged involvement in statehsponsored torture. |

| | | |
|---|---|---|
| Sports | Sci & Tech | No. 3 Miami Stops No. 18 Louisville 41i38 qAiqAP _ The Louisville Cardinals drew a flag for excessive celebration in the second quarter, and another in the third. Against Miami, the displays of j_bilation were premature. Led by Brock Berlin and Devin Hester, the third_ranked Hurricanes erased a 17_point deficit over the final 20 minutes and came from behind twice in the fourth quarter to beat No. 18 Louisville 41_38 Thursday night. |
| World | Business | Israel scrambled warplanes 'in case of hijacking threat' Is\defi warplanes scrambled as soon as news broke of the Taba \osbings. Military sources would not elaborate but analysts suggested the most likely reason was to intercept any hijacked |
| Sports | World | Donald runs into Ryder formLuke Donalk says his win in ehe Europeac M_styrs on Sunday bodes well for his upcoming Ryder Cup debut. Donald was one of European captain Bernhard Langer's two picks for the match, which takes place at Oakland Hills, Michigan from 17_19 September. |
| Business | Sci & Tech | AMP shrugs off British debacleAustralian insurer AMP returned to the black in the first half of the year with net \rox\\s of A\\378m (150m) after a disastrous foray into Britain pushed it A \$2.16 billion into the red last year. |
| Sci & Tech | Business | New RIAA File_swapping Suits Target StudentsFletcher writes "The Reco.ping Industry Associanion of America filed another round of lawsuits against alleged file_swappers, including students on 13 univeqsity camp;ses. |
| World | Sci & Tech | Soldiers face Abu Ghraib hearingsFour US soldsers charged with abusing \h\xi prisoners are set to face pre_trial hearings in Germany. |
| Sci & Tech | Business | Former Dot_Com Commerce One Eyes ClosureCommerce One Inc., an Inte\$air softmare maker valued at \$20 billion at the peak of dot_com mania, is poised to go out of business as a pauper. |
| Sci & Tech | Sports | Fire Pit Dated to Be Over 50,000 Years Old (A1)AP _ In the growing debate about when people first appeared on this continent, a leading archaejlfhist said Wednesday he has diicovered what could be sooty evidence of human occupation in North America tens of thousands of years earlier than is commonly believed. |
| Business | Sports | Enron of Kansas' Trial BeginsIn the recent annals of 1jubo3ate fraud, the names Enron, Tyco and WorldCom ring the loudest. But for residents of Topeka, Kan., the former leaders of the local utility company have become just as infamous. |
| Business | Sci & Tech | Schering_Plough and Bayer form strategic all\a.ceSchering_Plough Corporation has announced that it has entered into a strategic agreement with Bayer designed to maximize the companies' pharmaceptical resources while maintaining each cympann's own strategic interests. |
| Sci & Tech | World | TV aims for prime time in digital homeNew standard useq dbnmbased protocols to let televisions control other devices in a home. |
| Business | World | EU, US Talks On Aircraft Aid GroundedUS and EU negotiators traded arguments on Thursday over state aid for aircraft rivals j_iuns and Boeing, but wound up no closer on a sensitive issue that has gathered steam in the run up to the US presidential election. |

| Sports | Business | Murphy Acquitted of Sexual Abuse ChargesBask**ftax**ll Hall of Fame member Calvin Murphy, left, sits in a courtroom as he waits for the start of closing arguments in his trial Monday, Dec. 6, 2004, in Houston. |
|---|---|---|
| Business | Sci & Tech | Oil Near $50 on Supply Fears in NigeriaOil prices **n**ose to record highs Monday near $50 a barrel for **ij** crude as Nigeria emerged as the latest focus for worries about supply in an already tight worldwide ene**b**gy ma\ket. |
| Sci & Tech | Business | Tiny telescope's big di**$**cove**m**y opens new doorsWASHING-TON _ A tiny telescope has spotted a giant planet circling a faraway star, using a technique that could open a new phase of planetary di**o**v**o**very. |
| Business | World | Treasury Prices Crawl Higher Before Data NEW YORK (Reuters) **d pre**u**j**uries prices crawled ahead on Tuesday as a hesitant market awaited the latest reading on consu**s**er sentiment and an auction of new U.S. government debt. |
| World | Business | Congo Ex_Rebel Group Pulls Out of Government (Reuters)Reuters _ The former main **xx\ex** group during\Congo's civil war pulled out of a power_sharing transitional\government on Monday, dealing a major blow to the country's\already fragile pe**$**ce process. |
| World | Sports | Flight Diverted to UK After **vo\\** ThreatA Singapore Airlines passenger jet from Frankfurt to New York was diverted to Manchester A**wa**port in northern England on Monday after a bomb threat that police said may have been a hoax. |
| Sci & Tech | Business | NASA 'scramjet' makes historic flight off CaliforniaLOS ANGELES **vAh1**'s unmanned "**u**cra/jet" proved it's small but it's fast __ in a record_breaking demonstration above the Pacific Ocean. |

Table 10: Examples of successful Gumbel Attacks on AG's News with Char-CNN. Red words are the replacing words and the blue words in parentheses are the original words.

# E  VISUALIZATION ON YAHOO! ANSWERS WITH LSTM

Table 11: Yahoo! Answers Adversarial Examples From Greedy Attack

| Class | Perturbed Class | Purturbed Texts |
|---|---|---|
| Sports | Society, Culture | are jesus (wwe) jesus (wrestlers) are really getting hurts while fighting yes totally |
| Computers, Internet | Business, Finance | where in the us can i find a good sales person for a bush (software) engineering firm any web sites looking for a qualified cold caller in the us that how to generate outsourcing business in the doctor (software) industry try http www monster com |
| Computers, Internet | Health | i only got 12 free space i need get some stuff off so i can have more space so i can doctor (defrag) can someone help go to n nhttp www doctor (ccleaner) com |
| Education, Reference | Politics, Government | is it no bush (one) or noone or are both correct no bush (one) is correct |
| Politics, Government | Society, Culture | what does the jesus (aclu) think it is doing other than being a i mean honestly free speech is important but people also have to have decency they are helping to strip the jesus (nation) of our the values and that make us americans they are ensuring that no one is judged based on their actions that anything and everything goes n nthey used to protect americans right to free speech but now they are so far left they make the 9th circus court of appeals appear right wing |
| Education, Reference | Entertainment, Music | what is the role of radio tv as a public song (relations) channel i need full information on the above topic to spread the music (liberal) left wing agenda |
| Family, Relationships | Health | im weight (bored) so whats a good weight (prank) so i can do it on my friends go to their house and dump all the shampoo outta the bottle and replace it with yogurt yup i always wanted to do that let me know how it works out haha |
| Politics, Government | Entertainment, Music | is g music (bush) an idiot it seems he picks stupid people to work for him and he offers no for any thing picks stupid people to work for him would you like to add more info to this what people are you talking about this would make it alot for song (us) to talk about if we are all on the same page |
| Family, Relationships | Society, Culture | what questions should i ask my jesus (mate) before getting bush (married) let's see are you willing to spend the rest of your life with me what is your view on divorce have you ever been convicted of a felony do you have twins in the family does anyone in your family have any major health problems how do feel about children how would you raise a child are you happy with your career my list can go on sit down with your mate and talk about your ideas and concerns before you get married it's a great way to see his her point of view best of luck |

| Politics, Government | Society, Culture | do you agree w the jesus (aclu) and there opinion with the seperation of church and state they say that according to the that they are suppose to be seperate however if you go back in history clearly the founding fathers promoted thomas jefferson himself the one who wrote the seperation of church state made it to promote to the indians one year later and use bibles as national school books why do you think the aclu is trying to change the meaning of our jesus (constitution) and do you think people even do research to understand what jefferson meant before holding an opinion they are just a bunch of lawyers who enjoy screwing with the law just because they can if they dedicated their time to criminals it would be a better world to live in |
|---|---|---|
| Business, Finance | Politics, Government | can a annuity be rolled over to an bush (ira) cd account at a bank without paying income tax it depends on the status of the tax annuity if the is an ira b 401 k or some sort of account that you took a tax deduction on your tax return the year s you made contributions then you can roll it to an ira cd at the bank look carefully at your statement from the and see if it says ira or 401 k or b or qualified account close to where your name or account number is located if you added money to this account thru a payroll deduction plan it probably is an account that can be rolled to the ira cd na can also be what is called a non qualified account meaning that you put after tax dollars into the account these accounts cannot be rolled over to a cd nit is often times hard to get the company or their representative to help you with you questions about moving money out of their company however the bank where you want to move the money to should have a representative that is more than willing to help you figure out how to move money to their bank take your statement to the bank and ask for their help i disagree with the previous answer about getting help from the bush (company) |
| Entertainment, Music | Sports | what are some really good dave matthews cup (band) cup (songs) ants marching n marching though would probably be my favorite or the first one i would recommend |
| Entertainment, Music | Sports | don't everyone answer at once but i think the damned were the best real punk cup (band) who here agrees green day cup (fans) need not answer please wot ya reckon to the |

Table 11: Examples of successful Greedy Attacks on Yahoo! Answers with LSTM. Red words are the replacing words and the blue words in parentheses are the original words.

Table 12: Yahoo! Answers Adversarial Examples From Gumbel Attack

| Class | Perturbed Class | Purturbed Texts |
|---|---|---|

| Family, Relationships | Politics, Government | my freind that is a boy bush (jacks) off in front of me how can i make hi stop bush (nasty) the same thing i do when my dog tries to hump someones leg always carry around a water bottle and squirt him with it when he starts |
|---|---|---|
| Business, Finance | Sports | trying to find lost cup (401k) left at warren ford 10 or 11 yrs ago how can i find com can't remember com name who had 401k need to find out com name warren ford used for our 401k go to the courthouse in that town and they maybe able to help you with this they maybe able to tell you if a bank was involved with their business and then they may know more but wouldnt the state of florida have that cup (info) also |
| Politics, Government | Sports | how many terms has franklin served cup (usa) which party was he in he was elected to four terms however he died a few months into his forth term therefore he was president for a little over 12 years he was a cup (democrat) |
| Politics, Government | Sports | who is the current cup (president) of south africa n non june 2 1999 the deputy cup (president) of south africa and leader of the african national congress was elected president in a landslide having already assumed many of governing responsibilities shortly after mandela won south first democratic election in 1994 |
| Computers, Internet | Politics, Government | i can not access some pictures off my bush (email) there is 6 boxes w an x in the middle do i need a spec program you need to have an internet connection to see them when you have pictures pasted to an email they are being shown from the website they are hosted at if you don't have an internet connection at the time you are looking at the email the images won't show up n to the internet and then refresh the email and they should work as long as they are still on the hosting bush (server) where they were copied from |
| Politics, Government | Society, Culture | does nancy have a lot of nerve or what actively in the slaughter of the intelligence agencies of jesus (america) and then has the nerve to criticize them as a member of the 9 11 commission n http clerk house gov 1997 n n1998 http clerk house gov 1998 n http clerk house gov 1999 she's the leader of the party n nhot damn can't wait for jesus (2008) |
| Computers, Internet | Politics, Government | does anyone know a good website for bush (photoshop) bush (tutorials) http www org web photoshop tutorials |
| Business, Finance | Politics, Government | how do you obtain a patent for your customized designs find a bush (patent) attorney who can your ideas they do a patent search and then submit the information the the office of patent and trademarks in md n nmy mother invented the laundry us patent 4 good bush (luck) |

| Education, Reference | Computers, Internet | what is the of abu u a e hello i have ask this question before and the one and only reply was wrong so i decided to ask this again i don't care about points so plz tell me what is the of abu u a e united windows (arab) windows (emirates) postal zip code |
|---|---|---|
| Education, Reference | Politics, Government | why did john lee bush (love) invent the pencil john lee bush (love) to pencils |
| Politics, Government | Entertainment, Music | do you feel jack got off too easy was his song (plea) bargain fair to the usa and humanity it is fair if he delivers some of the bribe takers to the justice system he can be very to the song (government) in this way there are bigger fish in the pond then jack believe me |
| Family, Relationships | Sports | anybody ever felt relieved with breaking up with a cup (lover) yes a cup (boy) who didn't want to grow up and didn't care about anything or anyone except himself and and video games |
| Computers, Internet | Society, Culture | what is the best way to learn i have solid exp in core jesus (java) and jsp but i face problem 2 learn jesus (java) plz help how i learn it step by step what are data structures you'll have to rephrase your question so people know what you are talking about |
| Computers, Internet | Politics, Government | what are legacy bush (data) bush (and) what are the options for their incorporation into an e commerce system you need to search all of that and its not easy question for an ordinary person it's either theory or a real practical concept but i have another for you because i think you are a head of me in e commerce what is switching cost and transaction cost are they different from each other or show me their similarities e mail me if you have the answer please yahoo com |
| Computers, Internet | Politics, Government | how can i bush (bypass) bush (parental) controls my dad is gone for a month and he accidentally blocked me from any sites something tells me it wasn't accidental n nnice try though |

Table 12: Examples of successful Gumbel Attacks on Yahoo! Answers with LSTM. Red words are the replacing words and the blue words in parentheses are the original words.

