# OpenReview forum: "Greedy Attack and Gumbel Attack: Generating Adversarial Examples for Discrete Data"
_ICLR.cc/2019/Conference_

### Official Review · AnonReviewer2 · 2018-11-02
**Novel probabilistic framework for making adversarial attacks on deep networks with discrete valued inputs; flexible framework that allows solving the trade-off between attack success rate and computation time**

**Rating:** 7
**Confidence:** 4

**Review:**

The authors proposed a novel probabilistic framework to model adversarial attacks on deep networks with discrete inputs such as text. The proposed framework assumes a two step construction of an adversarial perturbation: 1) finding relevant features (or dimensions) to perturb (Eq. 3); 2) finding values to replace the features that are selected in step 1 (Eq. 4). The authors approximate some terms in these two equations to make the optimization easier. For example, it is *implicitly* assumed that given the i-th feature is removed from consideration, the probability of attack success does not change *on average* under probabilistic *adversarial* attack on other features (Eq. 5). It is not clear why that should hold and under what conditions that assumption would be reasonable (given that the attacks on other features are adversarial, although being probabilistic).
The proposed framework allows one to solve the computation vs. success rate trade-off by either estimating the best attack from the network (called greedy attack Eq. 6) or using a parametric estimation that does not require model evaluation (called Gumbel attack). Experimental results suggest that Gumbel attack has better or competitive attack rate on models developed for text classification while having the most computationally efficiency among other methods. It is also noticeable that the greedy attack achieves the best success rate with a large margin among all the tested methods.

---

> ### Author Response · Authors · 2018-11-15
> **Response to Reviewer 2**
>
> We thank the reviewer for the detailed and encouraging comments!
>
> To address the reviewer’s concern on Equation 5, we have added a more rigorous and detailed explanation of the approximation. Roughly, when one assumes other features are perturbed adversarially, the Greedy Attack can be interpreted as maximizing a lower bound of the original objectives. Details can be found in Section 3.1 of the updated version.

---

### Official Review · AnonReviewer3 · 2018-11-02
**Exciting advance in discrete adversarial attacks**

**Rating:** 8
**Confidence:** 2

**Review:**

In this work the authors introduce two new state-of-the-art adversarial attacks on discrete data based on a two-stage probabilistic process: the first step identifies key features which are then replaced in the second step through choices from a dictionary.

Overall the manuscript is very well written and easy to follow. The evaluation is extensive and contains all previous attacks I am aware of. The greedy attack outperforms all prior work by a large margin while the Gumbel attack works on par with the previous state-of-the-art while being significantly faster.

I only have a few questions and remarks:

* What’s the “random attack” baseline in these tasks? In computer vision it’s often sufficient to add a little bit of salt-and-pepper noise or Gaussian noise to change the model decision.

* Another thing I am wondering is what the human evaluation scores would be on adversarials from other adversarial attacks? Adversarial attacks in general (e.g. in computer vision) can work in two ways: one being actually changing the semantic content (thus also “fooling humans) while the other changes background features / add noise to which humans are pretty insensitive (unless you add too much of it). The greedy attack does seem to change some semantics as can be seen in the increased error rate of humans (which is pretty rare for computer vision adversarials). It might be that other attacks are rather changing words or characters which are not as semantically meaningful, as would be revealed by the accompanying human scores.

* Are you planning to release the code? Will it be part of CleverHans or Foolbox?

Overall, I find this work to be a really exciting advance on discrete adversarial attacks.

---

> ### Author Response · Authors · 2018-11-15
> **Response to Reviewer 3**
>
> We thank the reviewer for the encouraging comments and the help in addressing the importance of the task!
>
> What’s the “random attack” baseline in these tasks? In computer vision it’s often sufficient to add a little bit of salt-and-pepper noise or Gaussian noise to change the model decision.
>
> We define “random attack” as randomly sample k positions in the sentence, and replace them with randomly sampled words. We run random perturbation on the test set of the IMDB movie review dataset used in our paper. The average consistency of the predictions of the model from the perturbed and the original instances is 99.9% after k = 10 words are changed, and 92%, 90.4% after k = 50, 100 words are changed respectively. See the following link for a plot of comparison with our algorithms (on the first five words): https://drive.google.com/file/d/1T6UJQPz4iDFqsK9XQZ0nYv-bBcYxWraP/view?usp=sharing.
> We conclude that random perturbation does not work.
>
> “What the human evaluation scores would be on adversarials from other adversarial attacks?”
>
> We have added another experiment to compare various algorithms with human evaluation on the IMDB movie review data set. On each instance, we increase the number of words to be perturbed until the prediction of the model changes. Then we ask humans to label original texts and perturbed texts. Greedy attack yields the best performance in the experiment. Please see Appendix B of the updated version for details.
>
> “Are you planning to release the code? Will it be part of CleverHans or Foolbox?”
>
> Yes, we plan to release the code. We will either release the code in a stand-alone github repository or merge it into CleverHans.

---

### Official Review · AnonReviewer1 · 2018-11-02
**Important task; very poorly written**

**Rating:** 6
**Confidence:** 4

**Review:**

This paper addresses the problem of generating adversarial examples for discrete domains like text. They propose two simple techniques:
1) Greedy: two stage process- first stage involves finding the k words in the sentence/paragraph to perturb and second step changes the word in the positions identified in step 1.
2) Gumbel: first approach amortized over datasets where first and second steps are parametrized and learned over the dataset with the loss being the probability of flipping the decision.
Specifically, for the Gumbel approach, the authors use the non-differentiable top-k-argmax output to train the module in the second step which is not ideal and it would be better to train both first and second steps jointly in an end-to-end differentiable manner.

The results show that Greedy approach is able to significantly affect the accuracy of the systems compared to other adversarial baselines. Mturk evaluation shows that for tasks like sentiment analysis, humans weren't as confused as the systems were when the selected words were changed which is encouraging. However, the Gumbel method performs poorly compared to other baselines.
Moreover, a thorough analysis of why Greedy is doing better than some gradient based adversarial attacks is needed in the paper because it is unclear what is causing their greedy approach to perform well; is it the two-stage nature of the process?

My major gripe with the paper is that it is egregiously difficult to read in parts and is poorly written. There are dangling conditional bars in many equations (5, 7, Greedy attack etc.), unclear "expectation (E)" signs and many other confusing notational choices which make the math difficult to parse. I am not even sure if those equations are correctly conveying the idea they are meant to convey. I found  the algorithms to be more clearly written and realize that the text in the models and equations is unnecessarily complicated. The argument about approximation to the objective by considering the i positions independently is not convincing and their is nothing in the paper to show if the assumption is reasonable.

---

> ### Comment · Area_Chair1 · 2018-11-06
> **writing quality is extremely important**
>
> A poorly written manuscript is sufficient reason, by itself, to recommend rejecting a paper.
>
> Can you clarify how detrimental these writing problems are? Are they problems at the section and organizational level? The paragraph level in constructing clear prose? The sentence level? All of the above? Is the logical structure of the argument well-organized and easy to follow?

---

> > ### Comment · AnonReviewer1 · 2018-11-09
> > **Poor mathematical exposition**
> >
> > Organizationally, the paper is fine and sections are presented in a logical manner. But as I mentioned in my review, the mathematical exposition is certainly non-conventional and maybe even wrong. Their notations (Expectation symbols, conditional symbols etc.) have serious issues and I found their arguments about approximation and assumptions hard to follow. I'm not convinced about their argument of  approximating their proposed schemes by considering each position independently (partly because I can't clearly follow their argument) and moreover I believe that their original non-approximated probabilistic (unclear) formulation is unnecessary because it doesn't add anything to the paper.

---

> ### Comment · Area_Chair1 · 2018-11-07
> **Why is the task important?**
>
> Can you please clarify why you say the task is important? It is very easy to generate errors for models of text. Attackers would not need the methods in this paper to produce Yelp reviews that a state-of-the-art text sentiment classifier got wrong. They would not need any knowledge of machine learning at all to find errors for these text classifiers.

---

> > ### Public Comment · (anonymous) · 2018-11-09
> > **Indeed, why is this problem important?**
> >
> > Random perturbations are enough to fool text classifiers so why are the authors doing this? Because Gumbel-Softmax is fashionable?

---

> > ### Author Response · Authors · 2018-11-09
> > **Motivation**
> >
> > Dear Area Chair and Anonymous Reader:
> >
> > Thanks for your questions on the motivation of adversarial attack for discrete data. Below we briefly explain the motivation, followed by the evidence that simple random perturbation does not work.
> >
> > In summary, the area chair and another reader posed the following questions：
> >
> > 1. Why does one need to study the phenomenon of adversarial examples on discrete data?
> > 2. Why is this paper worth reading?
> > 3. Do simple methods like random perturbation work on text data?
> >
> > In short, our reply is
> >
> > 1. Robustness is an important criterion for models on discrete data. The generation of adversarial examples can be used to evaluate robustness or even improve robustness.
> > 2. In this paper, our goal is to propose methods with better performance (Greedy attack) or with higher efficiency (Gumbel attack).
> > 3. We provide evidence that simple methods like random perturbation do not work.
> >
> > Below are concrete details:
> >
> > Robustness is an important criterion for the application of machine learning models in critical areas such as medicine, financial markets, recommendation systems, and criminal justice. Adversarial examples have been used to evaluate the (adversarial) robustness of models (e.g., [1, 2, 5]) and have also been applied to train robust models (e.g., [3, 4]).
> >
> > The phenomenon of adversarial examples was first found in state-of-the-art deep neural network models for classifying images (e.g., [5, 6, 2]), where small perturbations unobservable by human can easily fool neural networks. Similar to image data, the problem of adversarial perturbation on discrete data can be defined as altering the prediction of a model via minimal perturbation to an original sample (e.g., [7-14]).
> >
> > While there have been many pioneered and interesting papers in this area (e.g., [7-14]), we proposed Greedy attack, a method to increase the misclassification rate of a model with a comparable scale of perturbation, and Gumbel attack, a method to improve the efficiency of generating adversarial examples, (It just happens to be fashionable :) ).
> >
> > It is natural to ask how the simplest algorithm, random perturbation, works before one is persuaded to read our paper. We compare our methods with random perturbation on the test set of the IMDB movie review dataset used in our paper. For each instance, we randomly sample k positions in the sentence, and replace them with randomly sampled words. The average consistency of the predictions of the model from the perturbed and the original instances is 99.9% after k = 10 words are changed, and 92%, 90.4% after k = 50, 100 words are changed respectively. See the following link for a plot of comparison: https://drive.google.com/file/d/1T6UJQPz4iDFqsK9XQZ0nYv-bBcYxWraP/view?usp=sharing.
> > We conclude that random perturbation does not work.
> >
> > [1] Carlini, Nicholas, and David Wagner. "Towards evaluating the robustness of neural networks." 2017 IEEE Symposium on Security and Privacy (SP). IEEE, 2017.
> > [2] Agarwal, Chirag, et al. "An Explainable Adversarial Robustness Metric for Deep Learning Neural Networks." arXiv preprint arXiv:1806.01477 (2018).
> > [3] Aleksander Madry, Aleksandar Makelov, Ludwig Schmidt, Dimitris Tsipras, and Adrian Vladu. Towards deep learning models resistant to adversarial attacks. ICLR (2018).
> > [4] Alex Kurakin, Ian Goodfellow, Samy Bengio. Adversarial machine learning at scale. ICLR 2017.
> > [5] Ian J Goodfellow, Jonathon Shlens, and Christian Szegedy. Explaining and harnessing adversarial examples. ICLR, 2015.
> > [6] Moosavi-Dezfooli, Seyed-Mohsen, Alhussein Fawzi, and Pascal Frossard. "Deepfool: a simple and accurate method to fool deep neural networks." CVPR, 2016.
> > [7] Ji Gao, Jack Lanchantin, Mary Lou Soffa, and Yanjun Qi. Black-box generation of adversarial text sequences to evade deep learning classifiers. IEEE Security and Privacy Workshops (SPW), 2018.
> > [8] Robin Jia and Percy Liang. Adversarial examples for evaluating reading comprehension systems. In Proceedings of the 2017 Conference on Empirical Methods in Natural Language Processing, pp. 2021–2031, 2017.
> > [9] Bin Liang, Hongcheng Li, Miaoqiang Su, Pan Bian, Xirong Li, and Wenchang Shi. IJCAI, 2018.
> > [10] Nicolas Papernot, Patrick McDaniel, Ananthram Swami, and Richard Harang. Crafting adversarial input sequences for recurrent neural networks. In Military Communications Conference, MILCOM 2016-2016 IEEE, 2016.
> > [11] Suranjana Samanta and Sameep Mehta. Towards crafting text adversarial samples. arXiv preprint arXiv:1707.02812, 2017.
> > [12] Minhao Cheng, Jinfeng Yi, Huan Zhang, Pin-Yu Chen, and Cho-Jui Hsieh. Seq2sick: Evaluating the robustness of sequence-to-sequence models with adversarial examples. arXiv preprint arXiv:1803.01128, 2018.
> > [13] Javid Ebrahimi, Anyi Rao, Daniel Lowd, Dejing Dou. Hotflip:White-box adversarial examples for text classification. ACL, 2018.
> > [14] Jiwei Li, Will Monroe, Dan Jurafsky. Understanding neural networks through representation erasure.  arXiv preprint arXiv:1612.08220, 2016.

---

> > > ### Comment · AnonReviewer3 · 2018-11-09
> > > **I second the importance**
> > >
> > > I'd like to second the importance of this work. Of course random perturbations at some point will also do the trick - but the same is true in computer vision applications where often small amounts of Gaussian noise lead to misclassifications. Nonetheless, many people in CV study adversarial perturbations as a means to understand what concepts network models have learnt and how susceptible they really are. Minimum adversarial perturbations are often several orders of magnitude smaller than random noise in CV, and the same seems to be true on discrete data like text.

---

> > > > ### Comment · Area_Chair1 · 2018-12-14
> > > > **I still don't understand your argument**
> > > >
> > > > Are you saying that finding nearby (in edit distance) errors has specific value? When do we need this in text?
> > > >
> > > > Are you saying that we should be surprised that the minimum perturbation in the Lp ball case is smaller than randomly found perturbations? If so, I disagree that this is surprising (see the thread above with Nicholas Carlini as well as the convincing paper he linked that supports this point). I also don't think this is surprising in the discrete case because the models considered certainly have a non-trivial amount of test error in the data distribution.

---

> > > > > ### Comment · AnonReviewer3 · 2018-12-14
> > > > > **of course minimum perturbations will be smaller than random perturbations**
> > > > >
> > > > > I did not mean to say that it's surprising that minimum adversarial perturbations are smaller than random perturbations. My comment was meant to second Nicholas in the importance of minimum adversarial perturbations. Just like him I'd like to ask you whether your arguments against the importance of this work would also hold for any paper on adversarials, whether on text or images? For image-based adversarial attacks and defenses there have been dozens of papers this year and the topic is important to many.

---

> > > > > > ### Public Comment · (anonymous) · 2018-12-14
> > > > > > **I don't understand either**
> > > > > >
> > > > > > I'm not reviewing this paper, but I'm interested in this discussion. It sounds like you are arguing that the topic of small worst-case perturbations is interesting because lots of people find it interesting. Can you be more specific? I'm not asking about adv ex as a whole, but more specifically papers which identify the minimum perturbation without much other contribution. What are we learning from the minimum perturbation, other than there are inputs for which the model makes a mistake and we found the nearest one? You are saying the distance of the minimum perturbation isn't surprising and I agree, so then what is the point of identifying the minimum perturbation?
> > > > > >
> > > > > > I'm all for an error analysis assuming we learn something from them. Maybe for text we could identify inherent biases of the model. For example, it's probably not a coincidence that in Table 3 modifying a word associated with sentiment (better) changed the model prediction. Just speculating but perhaps if the authors reran the random replacement experiment but with a bias towards inserting words associated with sentiment they would find random replacement would be more likely to degrade the performance of the model. What happens if you randomly append a bunch of sentiment related words as the last sentence of the paragraph? Identifying such a sentiment bias could be interesting and potentially useful.
> > > > > >
> > > > > > As a concrete example for computer vision, I really liked this paper which identifies a texture bias for computer vision models: https://openreview.net/pdf?id=Bygh9j09KX.

---

> > > > > > > ### Author Response · Authors · 2018-12-14
> > > > > > > **Your proposed experiments are interesting!**
> > > > > > >
> > > > > > > Wow! Thanks for proposing these interesting experiments! I personally agree with you that they are worth to investigate. Actually your thoughts point to some directions we are thinking about. I'd like to share with you some more of my personal thoughts, not necessarily related to the paper.
> > > > > > >
> > > > > > > You mentioned "papers which identify the minimum perturbation without much other contribution" may not be interesting. I agree with you. Experiments also need to be carried out to see whether humans can make the right decision after this, how to do this efficiently in the setting of adversarial examples, etc.
> > > > > > >
> > > > > > > As AC said: " this is not surprising in the discrete case because the models considered certainly have a non-trivial amount of test error in the data distribution." Yeah, that's truth! But does it suffice to know the test error exists? No, it is not. We need to investigate what the error is, or 'what is the inherent bias' as you mentioned, more importantly, 'how to characterize that bias?' Finally, 'how to fix that?'
> > > > > > >
> > > > > > > Fundamentally, the first thing we care about is: "How do we define the bias of a model?" I think maybe we can summarize in an inaccurate language: if most humans think the label of an instance is A, but the label is B, then the model has a bias. After that, we can proceed to find out "what is a precise way to summarize the bias of existing models". Maybe we can use math language, maybe we can summarize with other domain-specific abstraction (e.g.: texture). In the end, equipped with the knowledge, we proceed to fix the models. Perhaps one can use adversarial training? Or add some simple rules? So many potentially interesting directions there waiting for us!
> > > > > > >
> > > > > > > One more thing, although you pointed me to a nice paper, I feel sad about the cat in that paper who was enforced to wear the elephant skin:)

---

> > > ### Comment · Area_Chair1 · 2018-11-16
> > > **Thanks for elaborating**
> > >
> > > Thanks for elaborating on the motivation you have for the work. It is very helpful.

---

> > ### Public Comment · ~Nicholas_Carlini1 · 2018-11-09
> > **Motivating adversarial example research**
> >
> > [Disclaimer: I have not read the paper. This comment is solely intended to respond to the AC asking why the problem domain is important.]
> >
> > Adversarial example research papers have always had to deal with this question: why is this interesting? we already know classifiers make mistakes!
> >
> > There are at least a few common counter arguments:
> >
> > -  Yes, models make mistakes, but they are on average quite good. The interesting property of adversarial examples is that you can take an arbitrary input, that is very clearly Class A, and make the model produce the label for Class B. You can do this even when the object in Class A is the most A-like in the entire dataset. And even if class B resembles nothing like class A. That's what makes the domain interesting.
> >
> > - Why bother trying to find strong attacks if random noise might work? The main counter-argument here is that random noise often has to have a significantly larger distortion than adversarial noise. With Gaussian noise with sigma=0.2 on ImageNet, models still reach modest (50%+) accuracy. Adversarial noise with norm 20x smaller can reduce model accuracy to <1%.
> >
> > - Is this actually a security problem? It depends on the situation. For a nice treatment of this question see https://arxiv.org/abs/1807.06732

---

> > > ### Comment · Area_Chair1 · 2018-11-16
> > > **thanks for your comment**
> > >
> > > Thanks for weighing in, Nicolas, but I'm not sure I understand your argument.
> > >
> > > Neither of your first two points should surprise us when the models have substantial test error.
> > >
> > > To put it another way, 50% of random sigma=.2 perturbations are misclassified for ImageNet and adversarially chosen errors can be 20x closer than these randomly found errors. Of *course* the nearest error is going to be significantly closer than randomly found errors, the nearest error is, by construction, the nearest error! Why is 20x closer unusually close in a high (~150,000) dimensional space?

---

> > > > ### Public Comment · ~Nicholas_Carlini1 · 2018-11-16
> > > > **I agree with your perspective**
> > > >
> > > > There's actually a paper under submission that makes exactly this argument ( https://openreview.net/forum?id=S1xoy3CcYX ). I definitely agree that it should not be surprising that models have such low accuracy when you adversarially select noise to maximize classification error rate in light of this phenomenon.
> > > >
> > > > I don't think this is actually contradictory to adversarial examples as a line of research. In particular, one of the main reasons I see adversarial example work as interesting is that it gives us an estimate of the worst-case accuracy. Just like average-case accuracy is useful in many situations (and standard 'accuracy-on-test-set' measures do this for us), worst-case accuracy is also useful in other cases.
> > > >
> > > > To relate it back to this paper's topic of discrete data (again, I haven't read the paper) a classifier for malware that worked 100% of the time on "normal" data would be useless if it worked 0% of the time on adversarial data---because the only data it will ever see, malware, is by definition adversarial. The same argument applies to spam, and to a lesser extent various other written text attempting to avoid detection (e.g., the recent hate speech detectors).
> > > >
> > > > Just to clarify, though: it sounds like your "Why is the task important?" question is generally directed at the adversarial example research as a whole. Is this right? There are, by my count, at least 60 papers under submission to ICLR this year that focus explicitly on the problem of adversarial examples (explaining their existence, approaches for generating them, and approaches for defending against them). Would you have a similar complaint about any of these other papers?

---

> > ### Comment · AnonReviewer1 · 2018-11-09
> > **Adversarial examples help find systems' blind spots**
> >
> > The task is important because it focuses on small perturbations to text that change the classifier decision. These changes, if undetected by humans,  exhibit clear brittleness of a classifier and will aid in better design of a more robust classifier. This is the motivation for generating adversarial attacks in general: perturb the input ever so slightly in a manner that is in general undetectable to the human eye but results in drastic change in model's predictions. This has implications ranging from interpretablility and reliability of the model to security and privacy issues.
> > In the Mturk experiments, the authors do demonstrate that a lot of the perturbations produced by their models do not change the human's decision on sentiment classification but does change the model prediction.

---

> ### Author Response · Authors · 2018-11-15
> **Response to Reviewer 1**
>
> We thanks the reviewer for the comments, and the explanations on the motivation of the task. We have improved the clarity of Section 3.1 based on the reviewer’s suggestions. Below we respond in detail to the reviewer’s comments:
>
> “the Gumbel method performs poorly compared to other baselines”
>
> We agree that the performance of the Gumbel method is comparable to previous methods. However, its running time is significantly shorter than all the previous methods and our Greedy Attack method (See Figure 4). Thus, Gumbel Attack is the most efficient one across all methods even after taking into account the training stage. The efficiency of generating adversarial examples is an important factor for large-scale data.
>
> "what is causing their greedy approach to perform better” than “some gradient based adversarial attacks”?
>
> While gradient-based methods have led to several successful algorithms in the continuous domain (e.g., natural images), they have been observed to be less effective compared to discrete methods (e.g., [1]). It is mainly because gradient based methods focus on the sensitivity of response to each feature in the infinitesimal space, while perturbation is carried out in discrete space.
>
> “it is egregiously difficult to read in parts and is poorly written”
>
> We apologize for the difficulty of reading and have addressed the problem carefully. First, we have adopted the common notations and marked each expectation with a subscript to indicate the source of the expectation. Second, in the updated version, we have added a clearer and more detailed explanation of the approximation in Equation (5, 7).  To summarize, when one assumes other features are perturbed adversarially, the Greedy Attack can be interpreted as maximizing a lower bound of the original objectives.
>
> “The argument about approximation to the objective by considering the i positions independently is not convincing”
>
> We agree with the reviewer that this is an unnecessary assumption and have removed it from our framework (but still keep it in the design of Gumbel attack.) The independence assumption is used in Gumbel Attack for the sake of efficiency. This can be interpreted as a constraint on the search space so that decisions can be made in parallel. It can be a promising future direction to consider a framework where features are perturbed sequentially, with a termination gate [2] to control when to stop the perturbation. The latter enables the use of variable sizes of perturbation, instead of top-k perturbation.
>
> [1] Gao, Ji, et al. "Black-box Generation of Adversarial Text Sequences to Evade Deep Learning Classifiers." arXiv preprint arXiv:1801.04354 (2018)
> [2] Shen, Yelong, et al. "Reasonet: Learning to stop reading in machine comprehension." Proceedings of the 23rd ACM SIGKDD International Conference on Knowledge Discovery and Data Mining. ACM, 2017.

---

### Official Review · AnonReviewer4 · 2018-11-28

**Rating:** 3
**Confidence:** 4

**Review:**

This paper introduces two new methods for generating adversarial examples for text classification models. The paper is well written, the introduced algorithms and experiments are easy to understand.

However, I do not believe that these two methods are sufficiently significant. First of all, I am not convinced that the attacks can be classified as “adversarial examples”, especially the ones on the word-based models. The community originally got interested in adversarial examples because while they can easily be classified correctly by humans, they seemed to fool machine learning models with high efficiency. For example, the PGD attack by Madry et al. can reduce the accuracy of a CIFAR-10 model to 0% by using distortions that are not at all noticeable to humans. In the case of the word-based task studied here, human accuracy drops by 8-11%.

While the question of whether adversarial examples are actually a security threat is under debate, the attacks on the word-based models here do not even classify as adversarial examples. Of course, it is interesting that the ML models are much less robust to these distortions than humans are, however, this is a well known problem. This paper did not perform comprehensive experiments to investigate this phenomenon. For example, they could have evaluated a wide range of distortions (including random distortions), and then check if training with all of these distortions makes the network more robust … etc (for example, see [1]).

The attacks on character-based models are closer to adversarial examples from this perspective. However, the performance of the Gumbel Attack is significantly worse on character-based models than an attack as simple as the Delete-1 attack. The Greedy attack is more successful than the Delete-1 attack, however it is a straight-forward application of greedy optimization on discrete data and is not very novel or interesting.

[1] Generalisation in humans and deep neural networks, arXiv:1808.08750

---

> ### Author Response · Authors · 2018-12-01
> **Response to Reviewer 4**
>
> We thank the reviewer for reading our paper and giving detailed comments on our paper. However, we observe the review is posted after the deadline of paper modification has passed, and hope to address a few points of this review.
>
> In short,
> 1. We think the comparison between adversarial attack on images and on texts is unfair.
> 2. Some of the questions in the review are highly correlated with AC’s questions and we have answered in our previous rebuttal.
>
> We now address them in details.
>
> First, we agree with the reviewer that adversarial attack on texts is at a relatively new stage compared to the counterpart on images. However, to evaluate our work, we think it makes more sense to compare our methods with the best methods in this area, instead of with methods on a different data set. As an example, it does not make sense to reject a paper because their model achieves a lower accuracy on ImageNet than the accuracy of a very simple model on MNIST.
> We have shown that our method outperforms previous text adversarial attack algorithms in Figure 3, and we have even compared all methods under human evaluation in Appendix B, which indicates that humans are least sensitive to adversarial examples generated by our algorithm. So we believe our method can advance state-of-the-art in attacks on texts.
>
> “The attacks on character-based models are closer to adversarial examples from this perspective.”
> We aim to propose a general mathematical framework to generate adversarial examples for models with discrete input. Thus, the same algorithm works for both character-based and word-based models, and could be potentially useful for other NLP models such as word-piece models. We are happy to see that the reviewer consider character-based adversarial examples more interesting.
>
> “The Greedy attack is a straightforward application of greedy optimization on discrete data and is not very novel or interesting.”
> See our rebuttal to AC (point 2) in  “Efficiency of Gumbel Attack; Difference and Connections in Discrete Optimization and Adversarial Attack.” We will also elaborate this in our final version.
>
> The reviewer also proposed to include an experiment on how our attacks perform on models trained with data augmentation techniques. We agree with the reviewer that the proposed experiment can be interesting. However, given the timeline, we are not able to update the paper now. We are willing to add it in our final version.

---

> > ### Comment · AnonReviewer4 · 2018-12-06
> > **I'm then not convinced that "adversarial examples" in this context are interesting to study.**
> >
> > "First, we agree with the reviewer that adversarial attack on texts is at a relatively new stage compared to the counterpart on images."
> >
> > I brought this point up not because the methods introduced in this paper are not efficient enough at producing adversarial examples. I brought it up because I am not convinced that the distortions proposed in this paper, called adversarial examples by the authors, are significant enough for an ICLR acceptance.
> >
> > "As an example, it does not make sense to reject a paper because their model achieves a lower accuracy on ImageNet than the accuracy of a very simple model on MNIST. "
> >
> > I agree that these distortions on text data are not quantitatively comparable to similar distortions on image data. My concern is that finding distortions that fool text classifiers by itself is not a significant enough development. Almost all machine learning models fail to generalize to some small distortions ("small" as defined by distortions that would not fool humans). The authors present the distortions in this paper as especially worthy of study, as they fall under the category of "adversarial examples". I unfortunately do not find this convincing.

---

> > > ### Comment · AnonReviewer3 · 2018-12-08
> > > **Do the same arguments apply to vision?**
> > >
> > > Your arguments strike me as being equally applicable to adversarial images: there, a fairly small amount of salt & pepper noise is usually sufficient to fool DNN classifiers on ImageNet. Still, there are literally dozens of publications each conference looking at this problem. I fail to see why adversarials on text are less interesting than adversarials in the image domain.

---

> > > > ### Author Response · Authors · 2018-12-11
> > > > **We agree to disagree.**
> > > >
> > > > We express our sincere thanks to Reviewer 3 for the support. We think adversarial examples on texts are interesting to study, as is explained by Reviewer 3, as well as for the reasons we explained in previous posts.
> > > >
> > > > On the other hand, we also understand it is natural for different people to be excited about different areas, and to feel certain pieces of work less interesting. We still appreciate Reviewer 4 for reading our rebuttal.

---

### Author Response · Authors · 2018-11-15
**UPDATED: Elaborate Section 3.1 and add new human evaluation based on the reviewers’ suggestions.**

We have elaborated the Greedy attack with a clearer presentation in Section 3.1. First, we have adopted the common notations and marked each expectation with a subscript to indicate the source of the expectation. Second, in the updated version, we have added a detailed explanation of the approximation in Equation (5, 7).  To summarize, when one assumes other features are perturbed adversarially, the Greedy Attack can be interpreted as maximizing a lower bound of the original objectives.

We have added another experiment to compare various algorithms with human evaluation on the IMDB movie review data set. On each instance, we increase the number of words to be perturbed until the prediction of the model changes. Then we ask humans to label original texts and perturbed texts. Greedy attack yields the best performance in the experiment. Please see Appendix B of the updated version for details.

We again express our sincere thanks to all the reviewers, who have provided very useful suggestions for helping build our manuscript into a better shape.

---

### Comment · Area_Chair1 · 2018-11-16
**Can the reviewers please clarify the contribution(s)?**

As defined in this paper, an adversarial attack is just solving an optimization problem. For discrete sequence inputs, the paper considers a constrained discrete optimization problem. Discrete optimization is well studied and greedy algorithms for discrete optimization are also well-known and well-studied methods. They are obvious to machine learning practitioners as well. The particular greedy algorithm the authors use seems to be effective for this problem and does not require any special tricks.

Could the reviewers especially please comment on the following questions:

1. Is the Gumbel algorithm proposed necessary here or, more generally, is a new discrete optimization algorithm needed here?

2. The discussion section of the paper says:
"We have proposed a probabilistic framework for generating adversarial examples on discrete data, based on which we have derived two algorithms. Greedy Attack improves the state-of-the-art across several widely-used language models, and Gumbel Attack provides a scalable method for real-time generation of adversarial examples."

The paper claims to improve the state of the art. Can any of the reviewers comment on whether the paper advanced the state of the art in discrete optimization? Or, more generally, how should we read the claim above? Since a standard greedy algorithm works, there can't be anything special about this particular optimization problem that standard methods can't handle.

---

> ### Author Response · Authors · 2018-11-17
> **Efficiency of Gumbel Attack; Difference and Connections in Discrete Optimization and Adversarial Attack.**
>
> We thank the area chair for reading the reviews and our rebuttals carefully. We will answer the questions of Area Chair from the authors’ perspective.
>
> The area chair proposes two questions:
> 1) Why do we need Gumbel as a new discrete optimization algorithm?
> 2) Have we improved “the state-of-the-art” in discrete optimization?
>
> The short reply is
> 1)  Gumbel attack is efficient. The efficiency can be a practical concern in the setting of adversarial attack.
> 2) We propose better algorithms in terms of accuracy or efficiency in the regime of adversarial attack. This regime is not exactly the same as discrete optimization.
>
> We address the details below.
> 1)
> 1.1 Gumbel attack is efficient both in terms of the number of model evaluations and in terms of real time. First, no model evaluation is required during the attack stage. Also, Figure 4 in the manuscript provides a comparison of real-time efficiency, which shows Gumbel attack is orders-of-magnitude faster. (Gumbel attack is around 10^-2 seconds per sample while FGSM, Delete-1 Score and other methods are between 10^-1s and 1s per sample on Yahoo! Answers.)
> 1.2 In practice, attackers may not be able to conduct many model evaluations to attack a real system.
> 1.3 It may also help design more efficient adversarial training algorithms.
>
> 2)
> 2.1 We first address the difference between Greedy attack and standard greedy methods.
> The most standard greedy methods choose the first perturbation by evaluating models d * V times, where d is the length of the sentence/paragraph and V is the size of dictionary, and choose the next perturbation with complexity (d-1) *V, etc. Greedy attack follows a two-stage procedure motivated from a probabilistic framework, and takes O(d + k*V) evaluations in total (k being the number of perturbations). Moreover, Greedy attack is easier to parallelize. Given the efficiency concern of adversarial attacks, it can be more practical.
> 2.2 The area of adversarial attack is not exactly the same as discrete optimization.
> We formulate the problem of adversarial attack as a constrained discrete optimization problem. The true constraint here is that “humans will not change their decisions”, which we approximate by constraining the number of perturbed words.  Experiments involving human subjects have been carried out to validate the effectiveness of approximation.
> 2.3 We only show the superior performance of our algorithms to algorithms in adversarial attack ([1-4]), and we do not have the intention to claim it achieves the state-of-the-art in discrete optimization.
>
> [1] Ji Gao, Jack Lanchantin, Mary Lou Soffa, and Yanjun Qi. Black-box generation of adversarial text sequences to evade deep learning classifiers. arXiv preprint arXiv:1801.04354, 2018.
> [2] Jiwei Li, Will Monroe, and Dan Jurafsky. Understanding neural networks through representation erasure. arXiv preprint arXiv:1612.08220, 2016.
> [3] Nicolas Papernot, Patrick McDaniel, Ananthram Swami, and Richard Harang. Crafting adversarial input sequences for recurrent neural networks. In Military Communications Conference, MILCOM 2016-2016 IEEE, pp. 49–54. IEEE, 2016.
> [4] Bin Liang, Hongcheng Li, Miaoqiang Su, Pan Bian, Xirong Li, and Wenchang Shi. Deep text classification can be fooled. arXiv preprint arXiv:1704.08006, 2017.

---

### Author Response · Authors · 2018-12-15
**Why is the topic (finding nearby errors) important? (Part 1)**

1. Finding the error is the first step toward fixing it.

In general, given a sample x with correct prediction, the model is robust if it outputs the same correct label for all the nearby points *within a small distance*. The “defense” algorithms are proposed to improve the robustness of models, but before doing defense, it’s necessary to know *how to evaluate the robustness of a model*.

**There is no way to improve robustness if you don’t know how to measure it.**

Therefore, to evaluate the robustness of models, we need to *find nearby errors* for a given x. The AC might think doing random perturbation will work, but based on the experiments, simple random perturbation does not work for text (see our explanations in the “Motivation” threads) and also doesn’t work for image applications. Therefore, finding nearby errors is the first step before fixing it, so this has become an important task in our community. See the list of attacks for text data in our “Motivation” thread, and there’s much longer list of work on computer vision applications.

Moreover, recently researchers also found that *the error with small perturbation can be used to improve robustness*. The strategy is called “adversarial training”: When training the model, we can keep finding adversarial examples and adding them into training data to train the model. Random perturbation again doesn’t work well in this case, so the state-of-the-art method now works like 1) finds nearby adversarial samples based on current batch 2) run SGD on adversarial samples. See a seminal work (https://arxiv.org/abs/1412.6572) and one of the state-of-the-art methods (https://arxiv.org/abs/1706.06083). This is another reason that we want to find adversarial examples (nearby errors).

2. It’s not surprising that such nearby error exists. The question is how to find it.

We totally agree that it’s not surprising that such nearby error exists, and it’s also not our focus to show such nearby error exists. What we are doing in this paper is to propose an efficient way to find such error, which can be used to measure the robustness and identify the blind spots of the model (see our point 1).

3. Attacks can often be reduced to an optimization problem. Does this mean attacks are trivial?

As the AC pointed out and we also agreed, finding adversarial example (based on edit distance) for text classification can be formulated as a discrete optimization problem. Actually this is the case for most attacks. To attack a machine learning model, we want to

find x’ \in Ball(x, \epsilon) to *maximize* Loss(f(x’), y)

And this is naturally a constrained optimization problem. For image classification, x’ is in the continuous and bounded space. The seminal paper (https://arxiv.org/pdf/1312.6199.pdf) proposed to solve this by LBFGS. State-of-the-art  C&W (https://arxiv.org/abs/1608.04644) attack is based on the similar formulation with different loss functions. Given this, there are still tons of papers proposing different attacks in ICLR, in both black-box and white-box settings.

The AC pointed out that “Since a standard greedy algorithm works, there can't be anything special about this particular optimization problem that standard methods can't handle.” So does this imply there’s no contribution to apply an existing optimization algorithm to solve an ML problem? We disagree.

In general, we think “showing that an optimization algorithm works well for an ML problem” itself is an important contribution. There has been many works trying to apply existing optimization algorithms to ML models, such as SVM optimization, graphical models, sparse recovery, low-rank recovery, and these work have led to faster training/inference and many easy-to-use ML packages that were really beneficial to the community. Furthermore, when people apply some existing algorithm to some ML problem, they usually need to slightly change the algorithm to exploit the structure of problem, which is very important in practice.

We can see the same trend in the research of adversarial attacks, in both white-box setting (LBFGS is used initially, and then gradient descent, and then Adam), and black-box setting (coordinate descent is used initially, and then NES, genetic algorithm, etc). In our case, the algorithm is based on greedy optimization but has some treatments to make it adapt to text attack, see the discussion thread “Efficiency of Gumbel Attack; Difference and Connections in Discrete Optimization and Adversarial Attack.” Furthermore we show how to attack efficiently using the Gumbel trick, which is also an interesting finding.

---

### Author Response · Authors · 2018-12-15
**Why is the topic (finding nearby errors) important? (Part 2)**

4. In terms of security, finding the error is a way to attack the model.

From the security perspective, the ability of finding nearby errors (adversarial examples) leads to many security threats. Several important applications including attacking self-driving cars (slightly perturbed traffic sign can fool self-driving cars), and malicious ads (small change to ads pictures can pass the ML-based blocking or ranking models, see a nice new paper at https://arxiv.org/abs/1811.03194).

We believe this motivation is still valid for text data. For instance, if an attacker has a spam email to send out but the original email cannot bypass an ML-based spam filter, he or she can slightly perturb the text to bypass spam filter. The same thing can be done in many other online applications.

Or, for instance, A sends B a message or a document, and an attacker between A, B can change a small number of bits in the message to make the ML model in B mis-classify. Number of bits could correspond to the edit distance in text data, which is one motivation for using edit distance.

5. Is “edit distance” the best distance measurement for text adversarial example?

We think edit distance is a natural way as a distance measurement for text, but of course it might not be perfect. In general, even in image adversarial examples, there’s debate on which norm should be used.  FGSM ( https://arxiv.org/pdf/1412.6572.pdf) aims to control the L_\infty norm perturbation; C&W works for different L_p norm (https://arxiv.org/abs/1608.04644); recent papers also proposed more complicated norms such as L1-L2 norm (https://arxiv.org/abs/1709.04114), one pixel change (https://arxiv.org/abs/1710.08864), rotate/shift (https://arxiv.org/pdf/1712.02779.pdf) and semantic similarity (https://arxiv.org/pdf/1804.00499.pdf). We agree that exploiting different similarity measurement will be important, but in this paper we just focus on the most intuitive distance measurement in text and develop algorithms to find minimal adversarial perturbation. Also, we conduct human evaluation to show that minimizing edit distance works to some extent to achieve the goal that “the semantic meaning of the sentence is not changed”.

---

### Meta-Review · Area_Chair1 · 2018-12-14
**Although several reviewers support accepting this submission, I do not find their arguments for acceptance convincing**

**Confidence:** 5
**Recommendation:** Reject

**Metareview:**

I appreciate the willingness of the authors to engage in vigorous discussion about their paper. Although several reviewers support accepting this submission, I do not find their arguments for acceptance convincing. The paper considers automated methods for finding errors in text classification models. I believe it is valuable to study the errors our models make in order to understand when they work well and how to improve them. Crucially, in the later case, we should demonstrate how to use the errors we find to close the loop and create better models.

A paper about techniques to find errors for text models should make a sufficiently large contribution to be accepted. I view the following hypothetical contributions as the most salient in this specific case thus my decision reduces to determining if any of these conditions have been met. A paper need not achieve all of these things, any one of them would suffice:

1. Show that the errors found can be used to meaningfully improve the models.

This requires building a better model than the one probed by the method and convincingly demonstrating that it is superior in an important way that is relevant to the original goals of the application. Ideally it would also consider alternative, simpler ways to improve the models (e.g. making them larger).

2. Show that errors are difficult to find, but that the proposed method is nonetheless capable of finding errors and that the method is non-obvious to a researcher in the field.

This is not applicable here because errors are extremely easy to find on the test set and from labeling more data. If we demand an automated method, then the greedy algorithm does not qualify as sufficiently non-obvious and it seems to work fine, making the Gumbel method unnecessary.

3. Show that the particular specific errors found are qualitatively different from other errors in their implications and that they provide a unique and important insight.

I do not believe this submission attempts to show this type of contribution. One example of this type of paper would be a paper that does a comparative study of the errors that different models make and finds something interesting (potentially yielding a path to improved models).

4. Generate a new, more difficult/interesting, dataset by finding errors of one or more trained models

Given that the authors use human labelers to validate examples this is potentially another path. Here is an example of a paper using adversarial techniques in this way: https://arxiv.org/abs/1808.05326
However, I believe the paper would need to be rethought and rewritten to make this sort of contribution.


Ultimately, the authors and reviews supporting acceptance must explain the contribution succinctly and convincingly. The reviewers most strongly advocating for accepting this submission seem to be saying that there is a valuable new method and probabilistic framework proposed here for finding model errors. I believe researchers in the field could have easily come up with the greedy algorithm (a standard approach to discrete optimization problems) proposed here without needing to read the paper. Furthermore, I believe the other more complicated Gumbel algorithm proposed is not necessary given the similarly effective and simpler greedy algorithm. If the authors believe that the Gumbel algorithm provides application-relevant advantages over the greedy algorithm, then they should specify how these errors will be used and rewrite the paper to make the greedy algorithm a baseline. However, I do not believe the experimental results support this idea.